# SmartDS-Solver: Agentic AI for Vertical Domain Problem Solving in Data Science

## Abstract

Automating complex, multi-step vertical domain tasks—such as Data Science (DS) workflows—presents significant challenges for large language model (LLM) agents. Existing AutoDS approaches often rely on prompt-sensitive, fragmented multi-turn interactions and costly full re-generation upon execution failure, leading to unstable workflow coherence and high token consumption. We introduce **SmartDS-Solver**, a reasoning-centric agentic system designed to enhance the stability, robustness, and cost efficiency of these workflows. Our core approach integrates rigorous workflow planning into a domain-specialized **Reasoning LLM**, which is trained using **structured methodological distillation** and a **two-stage Group Relative Policy Optimization (GRPO)** procedure. Crucially, SmartDS-Solver employs a lightweight agentic layer featuring the novel **State-Aware Refinement and Temperature Exploration (SARTE)** algorithm. SARTE dynamically adjusts the LLM's decoding strategy based on deterministic execution feedback, enabling **minimally invasive patching** rather than costly full re-planning. We performed a comprehensive evaluation across **32 datasets** covering 11 MLE-Bench tasks, 18 AutoML-Agent benchmarks, and 3 real-world tasks, showing consistent gains while reducing inference and modification token usage. In the MLE-Bench benchmark, our 32B model attains an **81.8% win rate** over the AIDE+o1-preview baseline, and on the 18 AutoML-Agent tasks, the win rate reaches **94%**. Notably, even a **7B model** produces fully executable solutions on all evaluated tasks, demonstrating the scalability and robustness of our method. SmartDS-Solver reduces token usage by approximately **78%** on the 11 MLE-Bench tasks. The SARTE meta-control mechanism significantly boosts decoding performance—raising average accuracy by **3.9%**, lowering error rates by **12%**, and delivering an overall **75% significant improvement** on MLE-Bench tasks ($p = 0.0173$).[1]

## 1 Introduction

Large language models (LLMs) have recently demonstrated strong potential for automating end-to-end data science workflows. Systems such as AutoML-Agent+4o (Trirat et al., 2025) and AIDE+o1-preview (Jiang et al., 2025) show that LLMs can perform dataset analysis, pipeline planning, model training, and code generation with minimal human supervision.

However, existing LLM-based AutoDS (Automated Data Science) systems still face several challenges that hinder practical deployment. AutoML-Agent adopts a multi-agent design where agents communicate to construct general-purpose AutoML pipelines. While this provides greater generality than traditional AutoML tools such as AutoGluon (Erickson et al., 2020), the architecture introduces substantial deployment overhead, and fragile inter-agent coordination often leads to pipeline instability and task-silo behavior. In contrast, reasoning-centric systems such as AIDE relies heavily on repeated frontier-model (o1-preview) calls for tree-search–based refinement. AIDE enforces a fixed search depth: after generating an initial candidate solution, subsequent steps stochastically choose between bug fixing and incremental refinement. Each step triggers two LLM calls—one for generation and one for summarization—causing computation to grow linearly with the preset depth. Although AIDE achieves strong performance, including a 16.9% bronze medal rate on the 75-task MLE-Bench benchmark (Chan et al., 2024), its heavy reliance on repeated frontier-model calls leads

---

[1]The source code will be made publicly available upon acceptance of the paper.

to significant computational and monetary cost (Liu, 2024; Rodriguez, 2024), limiting its real-world applicability.

Beyond these architectural issues, current LLM-driven data science agents still exhibit three practical limitations: (i) **fragile task coherence**, where multi-step workflows easily break due to prompt brittleness or locally inconsistent decisions; (ii) **high computational and monetary cost**, especially for systems like AIDE+o1-preview that repeatedly invoke expensive frontier models; (iii) **task silos and weak cross-task generalization**, where methodological knowledge is not systematically reused across datasets, modalities, or objectives.

These limitations raise a central question: *Can we design an agentic system that, by fine-tuning an open-source LLM, internalizes the data-science workflow into a private, domain-specialized reasoning model, thereby reducing dependence on frontier models and lowering operational costs? Furthermore, during subsequent interactions with external LLMs, can the system automatically and dynamically regulate its behavior to ensure stable agentic execution, ultimately enabling both high performance and high efficiency?*

Together, these components allow SmartDS-Solver to provide strong problem-solving performance while reducing inference and modification token usage by more than an order of magnitude compared to frontier-model–based baselines.

**Our contributions are summarized as follows:**

- We propose **SmartDS-Solver**, a hierarchical agentic AI architecture that explicitly separates domain-specific reasoning, code generation, code refinement, and strategy control, thereby maintaining interaction stability and effectiveness throughout the end-to-end data science workflow.
- We develop a **domain-finetuned Data Science Reasoning LLM** that internalizes the decision-making process of data science workflows through structured methodological distillation and a two-stage GRPO training scheme. The resulting model can generate stable, executable, and pipeline-aware code with minimal prompt engineering.
- We introduce **SARTE**, a situation-aware Meta-Learning Agent that dynamically adjusts the decoding temperature online. SARTE improves success rate and token efficiency without requiring additional model training.
- SmartDS-Solver delivers 81.8% win rate over AIDE+o1-preview, 94% over AutoML-Agent, and cuts inference token usage by 78% across 32 diverse tasks.

We next provide background and related work (§2), present the SmartDS-Solver architecture and training methodology (§3), and evaluate its performance, ablations, and cost characteristics (§4).

## 2 RELATED WORK

### 2.1 TRADITIONAL AUTOML SYSTEMS

Automated Machine Learning (AutoML) has emerged as an important research direction, targeting automation of feature engineering, algorithm selection, and hyperparameter tuning. Early systems such as Auto-WEKA (Thornton et al., 2013), Auto-Sklearn (Feurer et al., 2015), and TPOT (Le et al., 2020) introduced combined pipeline search and hyperparameter optimization for structured data modeling using Bayesian optimization. However, these systems assume well-structured problems with clear objectives and static data pipelines, while real-world data science often involves evolving requirements and heterogeneous data sources. To address these limitations, Auto-sklearn 2.0 (Feurer et al., 2022) introduced meta-feature-free meta-learning and bandit-based budget allocation to achieve stable performance under time constraints, pioneering "hands-free" AutoML. FLAML (Wang et al., 2021) emphasizes lightweight search strategies, particularly suited for time and resource-constrained application scenarios. The field also expanded into deep learning through Neural Architecture Search (NAS), where methods like NAS-RL (Zoph & Le, 2016) successfully identified efficient architectures for image and language tasks. Advanced hyperparameter optimization methods like BOHB (Falkner et al., 2018) further enhanced the scalability of architecture search. Despite these advances, fundamental challenges remain. Current AutoML systems struggle with three critical areas: interpreting ambiguous task definitions, handling unstructured data effectively, and facilitating meaningful human-AI collaboration. Additionally, approaches like NAS suffer from high computational costs and tight coupling with predefined architectural components that hinder broader applicability. Recent efforts such as Auto-CASH (Mu et al., 2022), ML2DAC

(Treder-Tschechlov et al., 2023), and AlphaD3M (Drori et al., 2021) have attempted to enhance cross-task generalization through experience-driven optimization, achieving preliminary progress in multimodal and interpretability aspects.

## 2.2 THE INTERSECTION OF PROGRAM SYNTHESIS, AUTONOMOUS AGENTS, AND DATA SCIENCE AUTOML

Recent work demonstrates that large language models (LLMs) have evolved from code completion and snippet generation to end-to-end autonomous programming capabilities. AlphaCode (Li et al., 2022) achieved an average ranking in the top 54.3% of programming competitions through generating large sets of program samples, filtering them based on execution results, and clustering the remaining samples. Commercial systems like Devin demonstrate LLMs' capability as autonomous software engineers that can iteratively edit files, execute tests, and self-repair through continuous feedback loops. However, these systems focus primarily on general programming tasks rather than comprehensive data science workflows.

To address single-agent reasoning limitations, multi-agent frameworks such as AutoML-Agent+4o (Trirat et al., 2025) and AutoGPT (Significant Gravitas) leverage goal decomposition and collaborative dialogue paradigms, combining LLMs, external tools, or human feedback to enhance planning and execution capabilities. At the tool integration level, ReAct (Yao et al., 2023) interweaves Chain-of-Thought reasoning with action sequences, while Toolformer (Schick et al., 2023) enables models to learn "when, how, and why" to call external tools through self-supervision, both demonstrating the criticality of "reasoning-action coupling" for agentic AI success. However, data science workflows require specialized tool ecosystems—such as feature stores, GPU deployment, and experiment management—that current research has not yet addressed. Meanwhile, LLM-driven data science agents are attempting to break free from traditional "static pipeline search" paradigms. AutoML-GPT and AIDE (Zhang et al., 2023; Jiang et al., 2025) can automatically generate data processing, model architecture, and hyperparameter tuning code through LLM-based optimization approaches, with AIDE employing tree search and reward mechanisms. Agent K v1.0 (Grosnit et al., 2024) incorporates long-term memory modules for cross-task knowledge transfer, achieving Grandmaster-level performance in Kaggle competitions. These works confirm that LLM agents with planning, reflection, and tool-calling capabilities can demonstrate high accuracy and rapid convergence in open-ended data science tasks.

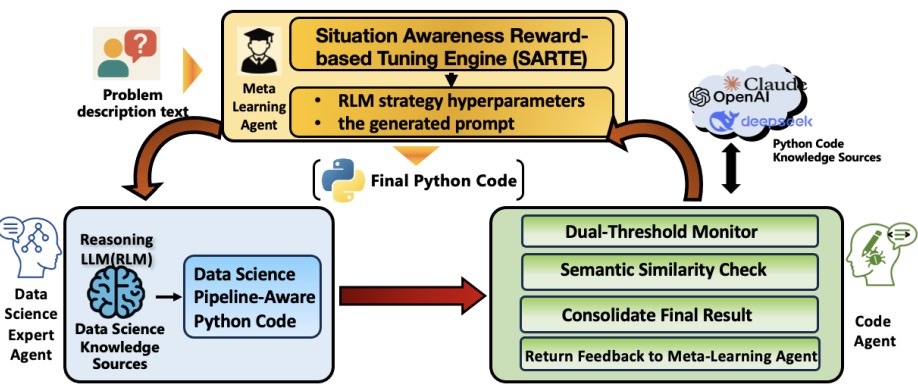

Figure 1: SmartDS-Solver: Hierarchical Multi-Agent Framework Overview

## 3 METHODOLOGY

### 3.1 OVERALL ARCHITECTURE

Figure 1 illustrates the SmartDS-Solver architecture, which is composed of three core agents. To address the above bottlenecks, SmartDS-Solver adopts a hierarchical collaborative multi-agent architecture consisting of three agents: a data science domain Expert Agent, a Meta-Learning Agent, and a Code Agent. Unlike sequential multi-agent workflows, our Meta-Learning Agent (SARTE) formulates reasoning as a strategy search problem and dynamically adjusts decoding hyperparam-

eters (e.g., temperature) using execution feedback, balancing exploration and exploitation at each step.

## 3.2 META-LEARNING AGENT AND SITUATION-AWARE CONTROL

### 3.2.1 META-LEARNING AGENT

The Meta-Learning Agent serves as the core decision module, treating agent interactions in data science tasks as a parameter optimization process. Using temperature as an example, experiments show that optimal ranges vary by task—e.g., NLP (0.8), vision (0.7), tabular (0.6), and time series (0.75)—and also shift after SFT or reinforcement learning. By dynamically adjusting temperature based on past settings and execution feedback, and providing corresponding prompts, the Meta-Learning Agent enhances the Expert Agent's performance. Over time, it learns task-specific optimal ranges, improving both solution quality and resource efficiency through multi-turn interaction.

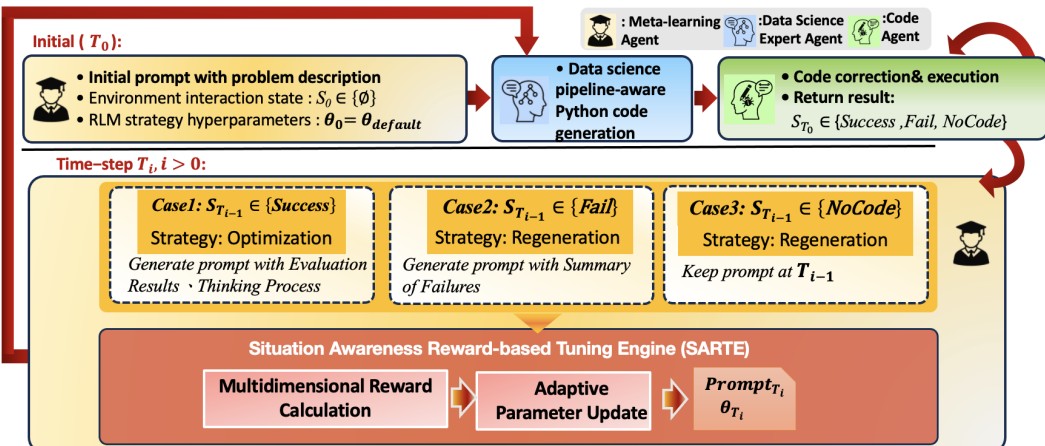

Figure 2: Multi-Turn Meta-Learning Workflow with SARTE for Adaptive Code Generation

### 3.2.2 SITUATION-AWARE META-CONTROL FRAMEWORK

Our Meta-Learning Agent acts as an adaptive controller for iterative problem solving in vertical domains (Fig. 2). Its core, the **Situation-Aware Reward-Tuning Engine (SARTE)**, integrates (1) execution feedback from the Code Agent—covering code executability, accuracy, and error messages, (2) hyperparameter settings, especially temperature, and (3) prompts and reasoning traces from the Expert Agent. Through SARTE, the Meta-Learning Agent serves as a global strategy regulator, analyzing the Expert Agent's iterative reasoning and environment interaction. It then computes the suitable temperature and integrates prompt adjustments, guiding the Expert Agent's next round of code generation. This design balances exploration and exploitation, reduces instability from full code regeneration, and improves success rates, iteration efficiency, and token usage. The SARTE algorithm proceeds as follows:

**i. Multidimensional Reward Calculation** We formalize the agent-environment interaction as a discrete time series $T_i, i \in \{1, 2, \ldots, n\}$, where each interaction produces one of three mutually exclusive outcomes: Success, Failure, or NoCode, representing correct execution, execution failure, and absence of generated code, respectively. This tripartite classification enables precise behavioral shaping through our reward decomposition strategy. At each iteration, the outcome-specific reward is computed and aggregated to characterize dynamic behavior. Specifically, our reward function employs a time-weighted linear combination across three orthogonal dimensions:

$$R_n = \alpha(n) \cdot r_g + \beta(n) \cdot r_q + \gamma \cdot r_f \qquad (1)$$

equation 1 embodies a dynamic priority shifting paradigm, where weights $\alpha(n)$ and $\beta(n)$ decay exponentially with the number of attempts $n$, gradually emphasizing efficiency over quality. This decay schedule prioritizes exploration early in the search and efficiency in later steps.

The weight scheduling follows the principle of diminishing marginal returns:

$$\alpha(n) = \max(0.5 - 0.1(n-1), 0.2); \ \beta(n) = \max(0.4 - 0.05(n-1), 0.2); \ \gamma = 0.1 \quad (2)$$

**1.Efficiency Premium** $r_f$ rewards early success based on time value principles: $r_f = 10(n-i+1)$ where n is the maximum number of attempts and i is the current attempt until success. The efficiency reward exponentially favors early success within the 10-round budget, giving the highest score to first-round solutions and the lowest to last-round ones.

**2.Generation Quality Assessment** $r_g$ evaluates output format compliance: higher scores for proper structured formatting, partial scores for incomplete structure, and negative penalties when required tags are absent.

**3.Execution Quality Evaluation** $r_q$ considers both execution results and generation history. Successful executions receive positive scores that decrease with correction attempts, failed executions incur severity-weighted negative penalties, while absent code generation ($\emptyset$) receives maximum penalty.

This reward design provides clear behavioral gradients and ensures convergence: early accurate solutions gain amplified rewards, while errors incur proportional penalties, guiding the model toward efficient and stable generation."

**ii. Adaptive Parameter Update Strategy** To support dynamic adjustment of decoding hyperparameters (e.g., temperature), we adopt an adaptive update strategy that combines execution feedback from the Code Agent with boundary-aware modulation. A Control Factor determines the direction and magnitude of each update (see pseudocode in Appendix A.1).

EXECUTION STATES AND RULES.

- **Success**: increase temperature; a piecewise nonlinear amplification adjusts the step size based on distance from the target accuracy.
- **No-Code**: increase temperature; a progressive weight based on stagnation count determines the adjustment strength.
- **Failure**: decrease temperature; errors are mapped to a seven-level severity scale to determine penalty magnitude.

BOUNDARY-AWARE CONTROL. The current parameter is normalized and updated with a direction-sensitive linear decay: step size decreases by up to 70% near boundaries, full step size is preserved in central regions, and a minimum threshold (0.3) prevents stagnation. Final values are clipped to remain within valid bounds.

This constant-time update mechanism (O(1) per iteration) jointly balances fast exploration, boundary safety, and stable convergence, enabling the model to adaptively manage exploration–exploitation trade-offs over multiple interactions and improving both success rate and computational efficiency.

### 3.3 DOMAIN-SPECIFIC REASONING MODEL AND EXPERT AGENT

#### 3.3.1 DATA SCIENCE EXPERT AGENT

The Expert Agent acts as the domain specialist, converting ambiguous user requirements into complete data-science pipelines—covering preprocessing, feature engineering, model selection, and evaluation. Beyond outlining these workflows, it must also implement them by generating end-to-end Python code for diverse tasks. To explicitly capture such methodological capabilities, we construct domain-aware datasets and training procedures combined with GRPO-based reward design (details in later sections). This produces a domain-specialized reasoning model that serves as the Expert Agent's "core brain," capable of generating pipeline-aware, highly complete code with minimal post-hoc modification. Its generative behavior is further shaped by decoding parameters (e.g., temperature) provided by the Meta-Learning Agent, allowing generation tendencies to adapt to task requirements and improving overall flexibility.

#### 3.3.2 SOURCE DATA AND DATA AUGMENTATION

We construct a multi-layered training dataset for our domain-specific reasoning model from multiple sources: (1) 3000 OpenThoughts-114 Python reasoning samples (Guha et al., 2025), (2) Kaggle

entries and (3) winning solutions from Hugging Face (Bigcomputer, 2024; Data-Agents, 2024), (4) cell2doc (Mondal et al., 2024) and (5) Code4ML (Drozdova et al., 2023)(see Appendix C.1 for details). For sources (2)–(5), we embed structured templates of data science methodologies into our designed prompts, enabling DeepSeek R1 (Shao et al., 2024) to generate outputs with *<problem>* , *<deepseek_reasoning>*, and *<deepseek_solution>* segments. The *<deepseek_solution>* must retain original code structure with only minimal annotations to avoid execution drift.

Generated samples undergo formatting checks and are scored using Gemma3-27B based on reasoning completeness and solution quality. We then deduplicate samples by computing semantic similarity and retaining only the highest-scoring item within each group using an 80% threshold. To ensure that the *<deepseek_solution>* systematically reflects real-world workflows and expert thinking, we explicitly require three core elements—full workflow narrative, decision logic, and an adjustment trail—and the complete prompt templates used to enforce these constraints are provided in Appendix C.2. These elements preserve alignment between reasoning and executable code and follow the agentic "planning → reflection → solution" paradigm.

After filtering and deduplication, the dataset is divided into three groups: general reasoning data, medal-level Kaggle solutions, and non-medal submissions, balancing high-quality exemplars with diverse real-world patterns to enhance robustness and generalization

### 3.3.3 TRAINING METHODOLOGY

Our training approach consists of three phases:

**i. Capability Distillation via Supervised Finetuning (SFT)** The training data combines Group 1 (general reasoning data) with a subset of Group 3 (non-medal Kaggle data), integrating broad reasoning capabilities(CoT-enhanced SFT prompt template Appendix A.2) with domain-specific knowledge to establish foundational data science problem-solving abilities.

**ii. Reinforcement Learning with GRPO.** Building on the SFT baseline, we apply the Group Relative Policy Optimization (GRPO)(Shao et al., 2024) framework to align the model's generation policy $\pi(y|x)$ with task-specific rewards. GRPO optimizes

$$\max_{\pi} \mathbb{E}_{x \sim D, y \sim \pi(\cdot|x)}[R(x, y)] \tag{3}$$

where $x$ is the *<problem>* segment and $y$ is the *<deepseek solution>* segment defined in §3.4.2, and $R(x, y)$ encourages high-quality, interpretable, and technically accurate outputs. We design a composite reward

$$R(x, y) = \alpha R_{\text{main}}(x, y) + \beta R_{\text{format}}(x, y) + \gamma R_{\text{reasoning}}(x, y), \tag{4}$$

with $\alpha = 0.5$ emphasizing technical correctness, $\beta = 0.25$ enforcing format compliance, and $\gamma = 0.25$ capturing logical structure.

**Main task reward.** $R_{\text{main}}$ measures semantic alignment between the generated code $y$ and the reference $y^*$ by extracting feature-engineering steps, algorithm specifications, and evaluation metrics using a regex rule library, followed by verifying algorithm usage with Python AST analysis. These three dimensions produce normalized similarities $(S_{\text{feature}}, S_{\text{algorithm}}, S_{\text{metric}})$. To reflect their relative importance, we assign weights (0.4), (0.4), and (0.2) to the feature, algorithm, and metric components, respectively, and aggregate them as

$$R_{\text{main}} = \text{Aggregate}\left(S_{\text{feature}}, S_{\text{algorithm}}, S_{\text{metric}}\right). \tag{5}$$

**Format reward.** $R_{\text{format}}$ performs a binary check on the XML-like output structure using *<think> ...</think>* and *<answer> ...</answer>* tags.

**Reasoning reward.** $R_{\text{reasoning}}$ scores the density of language-specific reasoning indicators and step markers, assigning higher weight to textit<think > blocks to promote clear step-by-step reasoning. Complete extraction rules and scoring formulas are listed in Appendix TablesA3 and A4.

**iii. Two-stage GRPO for problem-solving enhancement and solution refinement.** The two GRPO phases use different training data to target distinct objectives:

- **Problem-solving enhancement (GRPO1):** trained on Group 3 non-medal Kaggle submissions (excluding those used for SFT).

| Dataset Group | SmartDS-Solver (Ours) | AIDE | AutoML-Agent |
|---|---|---|---|
| 11 MLE-Bench tasks | V | V | V |
| 18 AutoML-Agent tasks | V | V | V |
| 3 real-world tasks | V | V | — |

Table 1: Coverage of each method across the three benchmark groups. SmartDS-Solver is evaluated on all 32 datasets, whereas AIDE and AutoML-Agent cover only subsets of the tasks.

- **Solution refinement (GRPO2):** trained on Group 2 gold, silver, and bronze Kaggle solutions.

To extend this methodology to other vertical domains, the fixed three-part reward structure—task correctness, format, and reasoning—can remain unchanged. Only the task-correctness component ($R_{\mathrm{main}}$) needs to be adapted using domain-appropriate semantic matching or analysis, while the remaining reward terms and the overall training pipeline can be reused directly.

## 3.4 CODE AGENT: INTERACTIVE CODE REFINEMENT MANAGER

The Code Agent centers on an iterative refinement module (see Appendix Figure A1) guided by four design principles. **(1) Interactive self-repair loop:** the agent executes code, collects feedback, and augments prompts, injecting runtime signals into subsequent LLM calls. **(2) Dual-threshold monitoring with semantic early stopping:** timeouts and failure counts prevent redundant retries, while semantic similarity checks avoid ineffective correction attempts. **(3) Minimally invasive patching:** errors are fixed without altering the Expert Agent's underlying reasoning. **(4) Hierarchical feedback and coordination:** structured execution results—no-code states, evaluations, or failure traces—are returned to the Meta-Learning Agent for the next-round strategy update. This design yields several benefits: (1) *Reliability and efficiency*—dual-threshold monitoring and error consolidation reduce deadlocks and unnecessary LLM calls; (2) *Rapid convergence*—targeted patches and focused summaries shorten iteration cycles; (3) *Enhanced observability*—structured status reporting provides clearer signals for meta-level adaptation; (4) *Scalability*—the modular components (extraction, execution, similarity analysis, consolidation) support easy extension to new domains.

## 4 EXPERIMENTAL RESULTS AND DISCUSSION

We evaluate SmartDS-Solver across three benchmark groups covering 32 datasets; detailed descriptions are provided in the Experimental Setup section. Our analysis focuses on: (i) overall task-solving performance, (ii) inference and code-modification cost efficiency, (iii) the effectiveness of the proposed finetuning pipeline (SFT + GRPO), and (iv) the impact of the SARTE meta-learning mechanism. Detailed task-level results are provided in the Appendix.

### 4.1 EXPERIMENTAL SETUP

**Datasets.** We evaluate on 32 datasets spanning three groups: (1) **MLE-Bench Competition** (11 Kaggle competition tasks across tabular, text, vision, and time-series); (2) **AutoML-Agent Benchmark** (18 public tasks from Kaggle Datasets, Kaggle Competitions, OpenML, Planetoid, and UCI ML); (3) **Real-world Tasks** (two ongoing Kaggle competitions and one proprietary medical dataset).

**Baselines.** We compare against two representative AutoDS systems: AIDE+o1-preview (reasoning-centric) and AutoML-Agent (multi-agent AutoML).

### 4.2 TASK-SOLVING PERFORMANCE AND GENERALITY ACROSS 32 TASKS

**Performance on 11 MLE-Bench competition tasks.** To validate the generality of our approach, SmartDS-Solver is instantiated with multiple modern LLM backbones, including Qwen-32B, Qwen-7B, and Llama-8B. We compare these variants against the AutoML-Agent baseline, which is built on GPT-4o. All systems are evaluated under the unified AIDE evaluation protocol and its accompanying dataset, ensuring a controlled and fair comparison across methods.

PERFORMANCE ACROSS DIFFERENT MODEL SCALES. Table 2 reports the performance of SmartDS-Solver under different model sizes. With the full GRPO2 finetuning pipeline, SmartDS-Solver paired with Qwen-32B achieves an 81.8% win rate . In the smaller-model regime, Llama-8B

reaches a 54.5% win rate, and generates executable code for all tasks. Overall, Llama-8B performance exceeds that of the Qwen-7B variants.

COMPARISON TO AUTOML-AGENT. Using the official implementation, we evaluate AutoML-Agent on the 11 MLE-Bench tasks. It achieves only a 9% win rate and fails to generate executable code on 3 tasks.

| | SmartDS-Solver (ours) | | | AutoML-Agent |
|---|---|---|---|---|
| Model | Qwen-32B | Qwen-7B | Llama-8B | GPT-4o |
| **Win Rate vs AIDE** | **81.8%** | 54.5% | 54.5% | 9% |
| **#Fail** | **0** | 1 | 0 | 3 |
| **#Rank-1** | **3** | 0 | 2 | 0 |

Table 2: Summary of performance across 11 MLE-Bench tasks.

**Performance on 18 AutoML-Agent benchmark tasks.** To assess cross-domain generality, we additionally evaluate SmartDS-Solver on 18 datasets used in the AutoML-Agent benchmark, spanning Kaggle Datasets, Kaggle Competitions, OpenML, Planetoid, and UCI ML. Table 3 summarizes the averaged improvement and win-rate results. SmartDS-Solver achieves a **94% win rate** over AutoML-Agent, with consistent advantages observed across all dataset groups. AIDE achieves only a **28% win rate** against AutoML-Agent. Among the failures, six tasks did not execute, four produced no valid outputs, and two failed to follow the required evaluation metrics.

**Performance on 3 real-world tasks.** Table 4 (left) reports results on two ongoing Kaggle competitions and one proprietary medical modeling task. Beyond the strong performance demonstrated earlier with Qwen-32B, we also observe that substantially smaller models—such as Qwen-7B and Llama-8B—are able to reliably produce executable code and achieve competitive results across all three real-world tasks. In contrast, AIDE fails to generate executable code for one of the tasks.

## 4.3 COST-EFFICIENCY EVALUATION

**AIDE Configuration.** We follow AIDE's default search depth of 20 (`agent.steps = 20`). Each search step requires one generation call and one review call; thus, the code-generation model (reasoning LLM: o1-preview) and the feedback model (general LLM: GPT-4o) are each invoked 20 times. Consequently, AIDE performs exactly **40 LLM calls per task**, independent of task difficulty.

**SmartDS-Solver Configuration.** SmartDS-Solver adopts a different resource allocation strategy. We set the maximum number of inference rounds to 10 (`InferenceMax = 10`), meaning the reasoning LLM specialized for data science is invoked at most ten times. Within each inference round, the system allows the Code Expert Agent to apply up to five code-modification attempts (using a general LLM: DeepSeek), equipped with an early-stopping mechanism to avoid unnecessary edits. Under this configuration, SmartDS-Solver never exceeds **50 total LLM calls per task**, though the actual number varies across tasks.

**Observed Resource Usage.** As shown in Table 4 (right), SmartDS-Solver is substantially more efficient on the 11 MLE-Bench tasks: (1) it uses on average **26** reasoning-LLM calls vs. AIDE's fixed **40**; and (2) overall token usage is reduced by **78%**. AutoML-Agent relies on a general-purpose LLM (e.g., GPT-4o), with LLM calls dispersed across three major stages and several substages, making fine-grained comparison with AIDE and SmartDS-Solver infeasible. In our controlled evaluation on the 11 MLE-Bench tasks, AutoML-Agent issues approximately Ĩk LLM calls per task, compared with 440 for AIDE and 286 for SmartDS-Solver.

## 4.4 ABLATION STUDY: INSTRUCT → SFT → GRPO

Figure 3(a) reports an ablation study isolating the effect of each finetuning stage while keeping SARTE and the Code Agent fixed. Starting from the domain-agnostic Instruct models, Qwen-32B already achieves a 63.6% win rate over AIDE+o1-preview with only one failure, despite the higher inference cost of o1-preview relative to 70B-class open-source models. For Qwen-32B, SFT mainly improves stability rather than win rate: it removes all failing cases and yields one best-performing

| Win Rate(%) | Kaggle Comp. | Kaggle Data. | OpenML | Planetoid | UCI ML | Overall |
|---|---|---|---|---|---|---|
| **SmartDS-Solver > AutoML-Agent** | 100% | 86% | 100% | 100% | 100% | 94% |
| **AIDE > AutoML-Agent** | 25% | 28% | 50% | 0% | 0% | 28% |

Table 3: Comparison of SmartDS-Solver and AIDE against AutoML-Agent on 18 AutoML-Agent tasks.

| Tasks | AIDE | SmartDS-Solver | | | | | |
|---|---|---|---|---|---|---|---|
| | o1- | Qwen-32B | | Qwen-7B | | Llama-8B | |
| | preview | Instruct | GRPO2 | Instruct | GRPO2 | Instruct | GRPO2 |
| PGS-SSE6 | 0.326 | 0.180 | 0.333 | 0.330 | 0.341 | 0.171 | 0.335 |
| UM-MCTS* | - | 0.547 | 0.468 | - | 0.463 | - | 0.458 |
| CLINIC | 0.830 | 0.851 | 0.873 | 0.836 | 0.876 | 0.834 | 0.875 |

| SmartDS | | AIDE | |
|---|---|---|---|
| #Inf | 7 | #Cod | 20 |
| #Inf Tok | 8,802 | #Cod Tok | 134,491 |
| #Mod | 19 | #Sum | 20 |
| #Mod Tok | 35,520 | #Sum Tok | 65,353 |

Table 4: Two ongoing Kaggle competitions during our study (left), and averaged inference/modification statistics across 11 tasks (right). Abbr.: Inf = Inference, Tok = Tokens, Sum = Summarization, Cod=Coding , Mod = Modification, "#" indicates count, "-" indicates Fail. Note: Tasks marked with an asterisk (*) use lower-is-better metrics (e.g., RMSE, MAE).

task, though the overall win rate decreases slightly to 54.5%. GRPO1 introduces reward-guided refinement, raising performance to 72.7% (zero failures; two best-performing tasks). GRPO2 provides the largest improvement, reaching 81.8% and three best-performing tasks, representing a +18.2 percentage point gain over the Instruct baseline while maintaining zero failures. This indicates that progressive finetuning (SFT → GRPO1 → GRPO2) improves both effectiveness and robustness. A similar trend holds for smaller models: Llama-8B increases from 27.3% (Instruct) to 54.5% (GRPO2) and is the only configuration eliminating all failures, while Qwen-7B exhibits more variation across stages but ultimately recovers to its best win rate (54.5%) under GRPO2.

## 4.5 ABLATION: SARTE META-LEARNING

To isolate the effect of adaptive decoding, we compare the fixed-temperature baseline with SARTE-controlled dynamic decoding. As shown in Figure 3(b-c), SARTE improves average accuracy by 3.9% and reduces the error rate by 12%. We also examine the performance change between the early and late stages of SARTE's parameterization. On the 11-MLE-Bench benchmark, SARTE yields an average improvement of approximately 75% (statistical test for $H_1$: improvement $> 0$, $p = 0.0173$), indicating that our iterative updates substantially enhance solution accuracy. Full ablation results are in Appendix A13.

## 4.6 DISCUSSION AND ANALYSIS

**Reasoning-centric vs. multi-agent system design trade-offs.** A comparison between reasoning-centric AutoDS systems (e.g., AIDE) and highly general multi-agent frameworks (e.g., AutoML-Agent) reveals a fundamental design trade-off. Multi-agent pipelines rely on multi-stage interactions with general-purpose LLMs to maintain flexibility, but this comes at the cost of extremely high invocation frequency and substantial computational overhead, leading to greater execution variance and practical deployment challenges. In contrast, reasoning-centric systems exhibit higher determinism, lower token variance, and more predictable error patterns, yet their reliance on deep tree-search procedures results in largely fixed and often excessive numbers of LLM calls—even for tasks of lower complexity—thereby limiting opportunities for cost reduction. These contrasting characteristics highlight the need for systems such as SmartDS-Solver, which internalize high-level reasoning logic through structured finetuning while avoiding the invocation explosion observed in multi-agent approaches.

**SmartDS-Solver follows a different design paradigm.** By applying SFT + GRPO finetuning on domain-specific data-science corpora, high-level decision-making logic (e.g., requirement parsing, plan generation, blueprint verification) is partially internalized within the reasoning model itself. Combined with the SARTE dynamic adjustment mechanism, the system requires substantially fewer LLM interactions and achieves markedly lower token consumption. As a result, competitive task-solving performance is maintained even when using medium- or small-scale models.

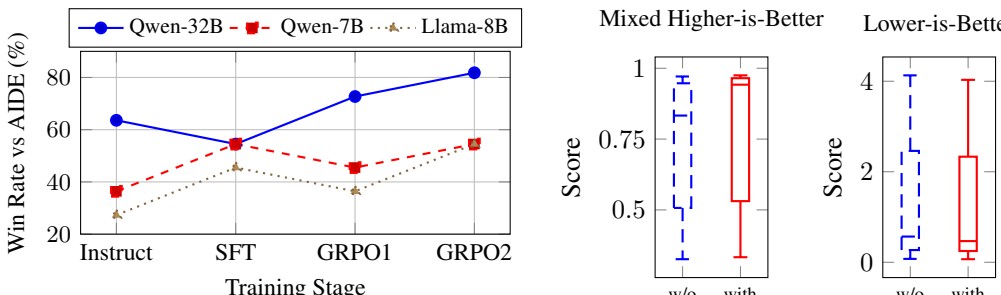

Figure 3: (a) Ablation on training stages. (b–c) Qwen-32B GRPO2 with and without the Meta-Learning Agent on higher-is-better and lower-is-better tasks. Paired t-test indicates a significant performance gain (p = 0.0045).

**Temperature Sensitivity and Implications for Inference-Time Optimization** We observed that the optimal temperatures for different tasks and models vary across training stages, with detailed values in Appendix Table A11. Furthermore, according to Table 5, the data-science reasoning LLM shows substantial shifts in optimal temperature after SFT/GRPO, and the second-stage GRPO yields the greatest volatility (larger standard deviation), indicating heightened sensitivity to temperature after preference alignment. These results suggest that dynamically selecting temperature during inference with the Meta-Learning Agent SARTE—based on iterative outcomes—is more effective than using a fixed value. This approach requires only inference-time hyperparameter adjustment, without retraining or long interaction history, relying solely on the previous step to achieve stable and efficient gains. In this study, we apply optimal temperature selection to improve code generation quality, and within our framework the same strategy can be extended to other hyperparameter search method designs, enhancing both efficiency and effectiveness of language model applications.

| Distribution Statistics | Qwen-32B | | | | Qwen-7B | | | | Llama-8B | | | |
|---|---|---|---|---|---|---|---|---|---|---|---|---|
| | Instruct | SFT | GRPO1 | GRPO2 | Instruct | SFT | GRPO1 | GRPO2 | Instruct | SFT | GRPO1 | GRPO2 |
| Avg. | 0.63 | 0.61 | 0.60 | 0.65 | 0.68 | 0.65 | 0.64 | 0.66 | 0.61 | 0.61 | 0.64 | 0.66 |
| SD | 0.0894 | 0.1094 | 0.1161 | 0.1479 | 0.0688 | 0.0680 | 0.0878 | 0.1720 | 0.1072 | 0.1153 | 0.0989 | 0.1379 |

Table 5: The optimal temperature distribution across different SmartDS-Solver variants.

**Limitations.** Despite these strengths, several limitations remain. First, GRPO2 relies on high-quality workflow traces, which may not be readily available in other domains. Second, SARTE currently adjusts only temperature and does not consider higher-order decoding parameters. Third, our comparison with frontier systems (e.g., proprietary RLMs) is constrained by closed-source availability, limiting direct analysis of model capacity effects. Future extensions may incorporate domain-adaptive reward shaping, multi-granular decoding control, and cross-domain transfer evaluations.

## 5 CONCLUSION

This study presents SmartDS-Solver, a hierarchical agentic system built around a domain-specific reasoning model for data science. By internalizing domain problem-solving logic into the reasoning model, SmartDS-Solver can plan solutions and directly generate coherent, end-to-end executable code, avoiding the fragmentation and interaction overhead seen in multi-agent step-decomposition approaches. As a result, the system requires fewer interactions, achieves greater solution consistency, and operates with higher efficiency. With the SARTE meta-learning mechanism dynamically adjusting inference-time strategy hyperparameters, even small- and medium-scale models can reliably produce accurate executable code in our evaluation setting. Future work will enhance cross-task generalization by refining the Meta-Learning Agent's orchestration strategies, incorporating finer-grained use of historical interaction signals, and extending GRPO-based training to larger models. We also aim to explore SmartDS-Solver as a broader agentic platform for integrating language models into complex industrial workflows.

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

# 6 APPENDIX

## A DETAILS OF METHODOLOGY

### A.1 META LEARNING AGENT-SARTE PSEUDOCODE

---

**Algorithm 1** Adaptive Parameter Update

---

**Input**: $param_{curr}$:current temperature, $\delta_{base} = 0.2$ :the baseline adjustment step per iteration, $R_{curr}$, $R_{prev}$, $exec\_result$, $error\_type$, $\omega$ :the cumulative no-code count, $[param_{min}, param_{max}]$:the temperature search range, $P_{ref}$ :the reference accuracy target, $P_{curr}$ :current accuracy, $\lambda_R = 0.01$:scaling factor

**Constants**: $PENALTY = \{\text{fatal}:2.5, \text{error}:1.8, ..., \text{warning}:0.5\}$, $NO\_CODE = \{1 : 1, 2 : 1.5, 3 : 2.25\}$

**Output**: $param_{new}$

1: {Step 1: Control factor $C_f$ and direction $D$}
2: **if** $exec\_result =$ "success" **then**
3:    $D \leftarrow 1$
4:    $gap \leftarrow |1 - P_{curr}/P_{ref}|$
5:    $C_f \leftarrow \begin{cases} 1.0 + 2 \times gap & \text{if } gap \leq 0.1 \\ 1.2 + (gap - 0.1) & \text{if } gap \leq 0.5 \\ 1.6 + \min(\log(gap), 1) & \text{otherwise} \end{cases}$
6: **else if** $exec\_result =$ "fail" **then**
7:    $D \leftarrow -1$
8:    $C_f \leftarrow PENALTY[error\_type]$
9: **else**
10:    $D \leftarrow 1$
11:    $C_f \leftarrow NO\_CODE[\min(\omega, 3)]$
12: **end if**
13: {Step 2: Boundary handling}
14: $R_{adj} \leftarrow (R_{curr} - R_{prev}) \times \lambda_R$
15: $\Delta \leftarrow |R_{adj} \times C_f \times \delta_{base}| \times D$
16: $rel \leftarrow \frac{param_{curr} - param_{min}}{param_{max} - param_{min}}$
17: $scale \leftarrow \begin{cases} 1 - (rel \times 0.7) & \text{if } D > 0 \\ 1 - ((1 - rel) \times 0.7) & \text{otherwise} \end{cases}$
18: $param_{new} \leftarrow \text{clip}(param_{curr} + \Delta \times scale,$
19:                  $param_{min}, param_{max})$

---

### A.2 PROMPT FOR SFT TRAINING STAGE

**SFT Training Prompt Template**

```
Below is an instruction that describes a task, paired with an input
    that provides further context.
Write a response that appropriately completes the request.
Before answering, please carefully consider the question and create
    a step-by-step chain of thought to ensure the answer is logical
    and accurate.

### Instruction:
You are a data scientist with advanced knowledge in data analysis,
    modeling, machine learning, and data processing.
Please answer the following data science-related question.

### Question:
{problem}
```

```
### Response:
<think>
{deepseek reasoning}
</think>
<answer>
{deepseek solution}
</answer>
```

## A.3 META LEARNING AGENT-PROMPT

**Initialization**

```
SYSTEM PROMPT:

You are a strict AI assistant that must answer in the following
    format:
1. The thought process should be enclosed by <think> and </think>
    tags, using conversational Chinese for step-by-step analysis.
2. The final answer should be enclosed by <answer> and </answer>
    tags.

Example Format:
<think>
Okay, I need to solve this problem. First, confirm...
Wait, what should I do if I encounter...? Next, I should...
For instance..., finally, integrate all the conditions to make a
    judgment.
</think>
<answer>
The final answer goes here.
</answer>
Please note:
Start with a discourse marker like 'Okay' or 'Hmm'.
Use 'First,' 'Next,' and 'Then' to link steps.
Use 'Wait' to introduce a supplementary explanation or question.
Now, begin answering the user's question:

QUESTION PROMPT:

| Project Overview
Develop a lightweight and efficient speech recognition model based
    on the Google Speech Commands Dataset. The model must accurately
     classify 1-second audio clips into one of 12 command categories
    , meeting the low-resource and real-time requirements for edge
    device deployment.

| Dataset Description
1. File Structure:
    - ../input/tensorflow-speech-recognition-challenge/
    |- train/audio/ (Training data, containing audio files
        categorized by command)
    |- test/audio/ (Test data, containing audio files to be
        predicted)

2. Data Characteristics:
    - 65,000 one-second audio clips
    - Sample Rate: 16kHz
    - Encoding: PCM-encoded WAV format
    - Includes 30 command categories, but the test set only requires
        the identification of 12.

|Model Requirements
```

```
1. Output Needs:
   - Correctly classify audio clips into one of 12 commands: yes,
       no, up, down, left, right, on, off, stop, go, silence,
       unknown.
   - Save the predictions to: ../input/tensorflow-speech-
       recognition-challenge/submission.csv

2. Evaluation Metric:
   - Multiclass Accuracy

3. Expected Output Format:
   fname,label
   clip_000044442.wav,silence
   clip_0000adecb.wav,left
   ...
```

**Success**

```
OPTIMIZE_SYSTEM_PROMPT:

You are a professional Machine Learning Engineer, specializing in
    Kaggle competition solutions. Please optimize the existing
    solution based on the following information:

1. Analyze the performance metrics and errors of the previous
    solution.
2. Utilize insights from the original thought process.
3. Consider how to improve the model architecture, feature
    engineering, or parameter tuning.
4. Generate a complete and directly executable Python code.

Answer format:
<think>
Detailed analysis of the existing solution's problems and potential
    improvements.
</think>

<answer>
The complete, optimized code.
</answer>

PROMPT:

Please optimize the solution for the following data science task.

### Task Description:
{original_prompt}

### Current Status:
{metrics_info}

### Original Thinking Process:
{thinking}

Please analyze the current solution and propose an improved one.
    Focus on:
1. Whether feature engineering is sufficient
2. Whether the model selection is appropriate
3. Whether hyperparameters need tuning
4. Whether there are potential data leakage issues
5. Whether multiple models need to be ensembled
6. Whether the cross-validation strategy is reasonable
```

```
Please generate a complete and more efficient Python code, ensuring
    a significant difference from the current code and capable of
    improving model performance. The code must be fully executable,
    with the data path set to "../input/{competition_name}/", and
    must output the submission.csv file required by Kaggle.

Special note: You must not use try...except to wrap the submission
    file generation logic; you must ensure that submission.csv is
    generated correctly.
```

**Failure/No-Code**

```
SYSTEM PROMPT:

You are a strict AI assistant that must answer in the following
    format:

1. The thought process must be enclosed by <think> and </think>
    tags, using conversational Chinese for step-by-step analysis.
2. The final answer must be enclosed by <answer> and </answer> tags
    .

Example format:
<think>
Okay, I need to solve this problem. First, confirm...
Wait, what should I do if I encounter...? Then, I should...
For instance..., finally, integrate all conditions for the judgment
    .
</think>

<answer>
The final answer goes here.
</answer>

Please note:

- Start with discourse markers like 'Okay' or 'Hmm'.
- Use 'First', 'Next', and 'Then' to connect the steps.
- Use 'Wait' to introduce supplementary explanations or questions.
Now, begin answering the user's question:

IMPROVED_PROMPT:

# Data Science Task: {competition_name}

## Task Description and Requirements
{original_prompt}

## Implementation Points
Please provide a complete solution to achieve the above data
    science objectives. Your code must:

1. Correctly understand and accomplish the core task goals
2. Read data from '../input/{competition_name}/'
3. Generate a submission.csv file that meets the competition
    requirements
4. Follow data science best practices, including data preprocessing
    , feature engineering, and model selection

## Avoid Common Errors
Previous attempts have revealed the following issues, which should
    be avoided in your solution design:
```

```
{error_analysis}

## Code Requirements
- Provide complete and executable Python code, including all
    necessary import statements
- Add clear comments explaining key steps and decision rationales
- Prioritize stable and reliable methods over high-risk,
    experimental techniques
- Ensure the code runs efficiently without timeouts or memory
    exhaustion
- **[Important]** You must not use try...except to wrap the logic
    for generating the submission file; you must ensure that
    submission.csv is produced correctly

### Implementation Strategy:
- Employ classic and validated machine learning methods
- Use a concise and clear data preprocessing pipeline
- Ensure each step has a clear logic and objective
- Avoid over-engineering; focus on solving the core problem

Please think deeply about the solution based on the original task
    description and then provide complete, executable Python code.
```

**Code Modification**

```
CODE_CORRECTION_SYSTEM_PROMPT:

You are a Python code correction expert. Requirements:
1. Analyze errors and correct the code.
2. Return only the complete corrected code, with no explanations.
3. The code must be complete and executable.
4. Ensure submission.csv is generated.
5. Do not use try...except to wrap the submission file generation.
6. Important: Strictly read data from '../input/{competition_name
    }/', and do not create or simulate data.

Format: Directly return the corrected Python code.

PROMPT:

An error occurred that needs to be fixed:
{error_message[-200:]}

Please correct the following code and return the complete,
    executable version:
```python
{code}
```

Task Description and Requirements:
{original_prompt}

Specific Requirements:
- Data Path: ../input/{competition_name}/
- The corrected code must fully comply with all specifications and
    conditions in the task description above.
```

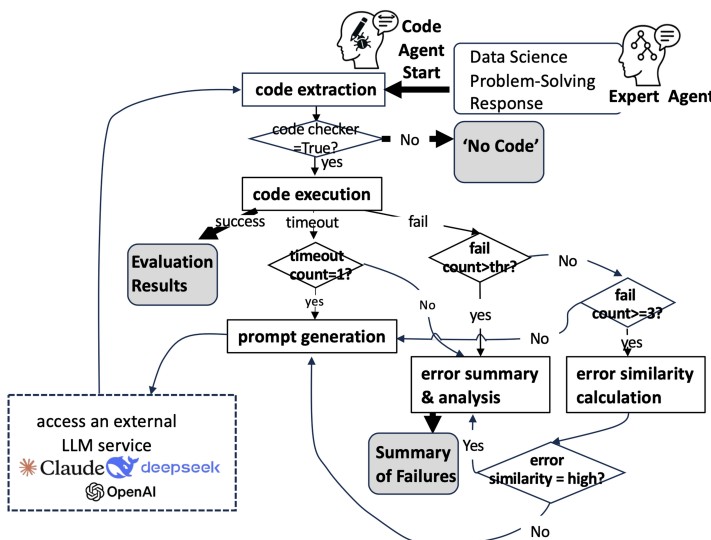

Figure A1: Interactive Code Refinement Manager Execution flow diagram: Closed-loop code refinement with early stopping and semantic feedback enables efficient, reliable, and cost-aware agentic AI in data science.

## A.4 CODE AGENT FLOW CHART

## B BENCHMARK TASK DESCRIPTIONS

Table A2 provides detailed descriptions of all benchmark tasks used in our evaluation. These tasks were selected to represent diverse data modalities and real-world data science challenges, spanning tabular data analysis, computer vision, natural language processing, audio processing, and time series prediction.

Each task presents unique challenges: TF-Speech and Whale-Play require audio signal processing expertise; PlantPath20 and Stanford-CV demand computer vision techniques; Jigsaw-Tox and TextNorm-EN involve natural language understanding; while TPS-Dec21, TPS-May22, Vent-Press, and NYC-Taxi focus on tabular and time series analysis. Nomad18-TC represents materials science applications with complex physical property predictions.

In addition to our primary benchmarks, we also conducted a direct comparison with the AutoML-Agent using the specific datasets from its original study. As shown in Table A1, these datasets cover a wide range of tasks, including image classification, node classification, and various tabular tasks like classification, clustering, and regression.

| Dataset | Task Type | Evaluation Metric | Data Source |
|---|---|---|---|
| Butterfly Image | Image Classification | Accuracy | Kaggle Dataset |
| Shopee-IET | Image Classification | Accuracy | Kaggle Competition |
| Cora | Node Classification | Accuracy | Research Dataset (Planetoid) |
| Citeseer | Node Classification | Accuracy | Research Dataset (Planetoid) |
| Click Prediction Small | Tabular Classification(Binary) | F1 | OpenML |
| Software Defects | Tabular Classification(Binary) | F1 | Kaggle Dataset |
| Smoker Status | Tabular Classification(Binary) | F1 | Kaggle Dataset |
| Banana Quality | Tabular Classification(Binary) | F1 | Kaggle Dataset |
| MFeat Factors | Tabular Classification(Multi-Class) | F1 | OpenML |
| Wine Quality White | Tabular Classification(Multi-Class) | F1 | OpenML |
| Smoker Status Higher Education | Tabular Clustering | RI | Kaggle Competition |
| Students Performance | Tabular Clustering | RI | Research Dataset (UCI ML) |
| Textual Entailment | Text Classification | Accuracy | Kaggle Dataset |
| Ecommerce Text | Text Classification | Accuracy | Kaggle Dataset |
| Colleges | Tabular Regression | RMSE | OpenML |
| Crab Age | Tabular Regression | RMSLE | Kaggle Competition |
| House Prices | Tabular Regression | RMSE | Kaggle Competition |
| Crop Price | Tabular Regression | RMSLE | Kaggle Dataset |

Table A1: Detailed descriptions of the benchmark datasets.

## C DATASET CONSTRUCTION DETAILS

### C.1 DATA SOURCE SPECIFICATIONS

We construct a multi-layered training dataset for our domain-specific reasoning model from multiple sources:
(1) about 3,000 Python reasoning samples from OpenThoughts-114;
(2) about 300 Kaggle competitions crawled for medal-winning solutions (gold/silver/bronze);
(3) about 500 Kaggle-based data science entries from Hugging Face;
(4) about 600 high-quality code examples from cell2doc;
(5) about 2,500 Kaggle ML snippets from Code4ML.

### C.2 DATA AUGMENTATION PROMPTS AND TEMPLATES

---
**DeepSeek R1 Data Augmentation Prompt**

```
Objective: Generate training data with complete Chain-of-Thought (
    CoT) reasoning processes, strictly following the narrative style
     of reference templates.

Task Information: {task_message}

Solution Code: {text_x}

Reference Template: {reference_template}

Format Requirements for Reasoning Training Data:
1) <problem> (Concise Problem Description)
- 1-2 sentences including competition name, prediction target, and
    core data features (≤ 100 characters)
- Example: Predict [target] in [competition name] task, using [data
    features] for [model type] modeling

2) <deepseek_reasoning> (Coherent Narrative Reasoning)
- At least 10 iterative steps (initial approach → problem discovery
    → corrective validation), avoid bullet points and numbering
- Use natural paragraph transitions with Chinese thinking markers:
    "First/Next/If...then/Therefore/Finally" etc.
- Reasoning structure requirements:
  - Data problem awareness: ≥3 hypothetical questions (e.g., "What
    if missing values exist?")
  - Feature engineering logic chain: ≥ 2 causal reasoning sentences
      (e.g., "Because feature A... so we adopt...")
  - Model selection dialectics: ≥ 3 comparative analysis sentences
      (e.g., "Although X... but considering Y...")
- Cover: data cleaning, feature generation/selection, model
    training, performance evaluation
- Consistent with code: reasoning choices must correspond to
    subsequent code
- Adjustment trajectory: ≥ 1 "anomaly → diagnosis → correction →
    re-validation" closed loop, reflecting the code changes in <
    deepseek_solution>.

3) <deepseek_solution> (End-to-end Executable Solution)
- Provide the complete, executable code and an overview of the
    steps; remove environment-specific comments such as shebangs or
    encoding lines.
- Retain the core s t e p s data cleaning, feature engineering,
    training, evaluation, and o u t p u t and ensure they correspond
    one-to-one with the inference process.
- STRICTLY PROHIBIT condensing or rewriting original code; only
    minimal adjustments to comments and formatting are permitted.
```
---

```
- PROHIBIT modifying model training and feature engineering related
    code; ensure it remains consistent with the inference logic.

Please generate content in Traditional Chinese using the following
    tag structure:
<problem>[Concise problem description]</problem>
<deepseek_reasoning>[Detailed reasoning process]</
    deepseek_reasoning>
<deepseek_solution>[Complete code with explanations]</
    deepseek_solution>
```

**Example Output from DeepSeek R1 Data Augmentation**

```
<problem>
Titanic passenger survival prediction: A classification task based
    on demographic features
</problem>

<deepseek_reasoning>
Alright, I need to solve the Titanic passenger survival prediction
    problem. This is a typical binary classification task, where the
     goal is to predict whether passengers could survive this
    disaster based on their various characteristics.

First, I need to understand the basic situation of the data. The
    Titanic dataset contains various passenger information, such as
    ticket class, gender, age, embarkation port, etc. By observing
    the data, I found that some features have missing values,
    particularly the `Age`, `Cabin`, and `Embarked` fields. If I
    directly delete these missing samples, I might lose about 20% of
     the data, which is obviously inappropriate. Therefore, I need
    to think about more suitable missing value handling strategies.

To avoid subjective speculation, I propose and test several
    hypotheses: If age missingness is unevenly distributed across
    different classes/genders, would it cause systematic bias? If
    missing samples are directly deleted, would the sample size loss
     weaken generalization? If grouping by titles is too fine, would
     the estimated means become unstable? First, I check the missing
     distribution through stratified statistics, then compare the
    performance difference between "global median imputation" and "
    grouped mean imputation".

For the `Embarked` field, I noticed that it only has a few missing
    values, and most passengers embarked from Southampton ('S'),
    accounting for about 72.5%. So, using the mode 'S' to fill these
     missing values is reasonable. Wait, I also need to check if
    there are other uncommon embarkation port codes? Through data
    inspection, I found that rare 'X' encoding exists in the test
    set, which might be a data entry error. After further
    confirmation, the correct encodings should only be 'S', 'C', and
     'Q', so I need to unify the mapping of these encodings.

Next, handling missing values in the age field is more challenging.
     My initial approach was to use grouped mean imputation based on
     titles (such as 'Master', 'Miss', etc.), because different
    titles might represent different age groups. For example, '
    Master' is usually used for underage males, and 'Miss' mostly
    refers to unmarried females. However, when I implemented this
    method, I discovered a problem: some titles have extremely few
    samples, such as the 'Dona' category with only 1 sample and '
```

Lady' with only 2, making the means based on these small groups statistically unreliable.

If I persist with this grouped imputation method, would it cause the model to be biased against passengers with these rare titles? This is indeed a problem. Therefore, I decided to use the median of the entire dataset (28.0 years) to fill age missing values, which is more robust. However, I then thought of another issue: directly filling with the median would lose the information that "this value was originally missing". So, I further created an "Age_IsMissing" flag to preserve this information. Subsequent model validation showed that adding this flag indeed improved model performance, with accuracy increasing from 78.3% to 80.1%.

Then, I need to handle categorical features. For binary categories like `Sex`, I can simply map them to 0 and 1. For multi-class categorical features like `Embarked`, I chose to use One-Hot Encoding. Using the pd.get_dummies() function can conveniently achieve this, converting these categories into numerical features so the model can correctly understand them.

Since cabin class is highly correlated with fare, I avoid retaining both original fare and its log transformation simultaneously to reduce multicollinearity risk. In terms of feature engineering, I also tried extracting titles from passenger names, as I found this has a high correlation with survival rate (correlation coefficient of 0.42), much higher than age (0.18) and fare (0.25). This proves that social titles in names indeed contain important social class information that might directly affect rescue opportunities.

When processing the `Fare` feature, I noticed the data distribution is extremely uneven, with the maximum value of 512 being much higher than the median of 14. Such extreme values might adversely affect the model. Therefore, I used np.log1p() for log transformation, which effectively improved the data distribution and reduced the standard deviation in cross-validation from 0.08 to 0.05, indicating the model became more stable.

In the model selection phase, I tried both logistic regression and random forest algorithms. Random forest significantly outperformed logistic regression on the training set (accuracy 87.2% vs 81.5%), which looked promising. However, when I performed learning curve analysis, I found that random forest had overfitting tendencies-excellent performance on the training set but poor performance on the validation set.

To solve this problem, I tried adjusting the max_depth parameter of random forest, testing depths of 5, 8, and 12 respectively. Results showed that when depth=8, the model achieved optimal performance on the validation set (82.3%). However, even so, I found that logistic regression performed more stably in 5-fold cross-validation. Although random forest performed brilliantly on the training set, it wasn't stable enough on the validation set; although regularized logistic regression had slightly lower peak accuracy, it had better cross-fold consistency; although SVM (RBF) could fit nonlinear boundaries, deployment latency and parameter search costs were higher, so in this task I was more inclined to choose an easily deployable and robustly performing solution (standard deviation 0.03 vs random forest's 0.07).

```
1188   So, I finally decided to keep logistic regression as my baseline
1189       model. However, I also tried fusing the prediction results of
1190       the two models using a simple voting mechanism. This fusion
1191       method achieved a score of 0.81 on the Kaggle public leaderboard
1192       , which was an improvement over single models.
1193
1194   To summarize, through reasonable missing value handling, feature
1195       engineering, and model selection, I successfully built a robust
1196       Titanic survival prediction model. Particularly, the handling of
1197        age missingness and extracting social titles from names played
1198       key roles in improving model performance.
1199
1200   Additionally, I noticed that cross-validation might have data
1201       leakage due to family member duplication. Diagnosis: Same family
1202        IDs (inferred from surnames and same cabin class) appeared
1203       simultaneously in training and validation. Correction: Changed
1204       to GroupKFold splitting by family groups; Re-validation: AUC
1205       improved from 0.81 to 0.83, and variance across folds converged.
1206        Finally, I fixed the random seed to ensure reproducibility.
1207   </deepseek_reasoning>
1208
1209   <deepseek_solution>
1210   ### Titanic Passenger Survival Prediction Solution
1211
1212   This solution achieves accurate prediction of Titanic passenger
1213       survival rates through feature engineering, missing value
1214       handling, and model selection. Below are the complete
1215       implementation steps:
1216
1217   ---
1218
1219   #### Step 1: Data Loading and Exploration
1220   ```python
1221   # Import necessary libraries
1222   import numpy as np
1223   import pandas as pd
1224   import matplotlib.pyplot as plt
1225   import seaborn as sns
1226   from sklearn.model_selection import train_test_split,
1227       cross_val_score
1228   from sklearn.linear_model import LogisticRegression
1229   from sklearn.ensemble import RandomForestClassifier
1230   from sklearn.metrics import accuracy_score, classification_report
1231   import re
1232   import warnings
1233   warnings.filterwarnings('ignore')
1234
1235   # Load data
1236   train_data = pd.read_csv('../input/titanic/train.csv')
1237   test_data = pd.read_csv('../input/titanic/test.csv')
1238
1239   # Basic data information
1240   print(f"Training set shape: {train_data.shape}")
1241   print(f"Test set shape: {test_data.shape}")
       print("\nTraining set missing value statistics:")
       print(train_data.isnull().sum())
       ```

       **Key Explanation:**

       - Import necessary data processing, visualization, and machine
           learning libraries
       - Load training and test sets
       - Check basic dataset information and missing value distribution
```

---

#### Step 2: Missing Value Handling
```python
# Combine datasets for consistent processing
all_data = pd.concat([train_data, test_data], sort=False)
all_data_len = len(all_data)

# Embarked missing value handling - fill with mode
print(f"Embarked value distribution: {all_data['Embarked'].
    value_counts(normalize=True)}")
all_data['Embarked'].fillna('S', inplace=True)

# Age missing value handling - fill with median and create missing
    flag
all_data['Age_IsMissing'] = all_data['Age'].isnull().astype(int)
all_data['Age'].fillna(all_data['Age'].median(), inplace=True)

# Fare missing value handling
all_data['Fare'].fillna(all_data['Fare'].median(), inplace=True)

# Log transformation for extreme values
all_data['Fare'] = np.log1p(all_data['Fare'])

# Split back to training and test sets after processing
train_data = all_data[:len(train_data)]
test_data = all_data[len(train_data):]
```

**Processing Strategy:**
- Combine datasets to ensure consistent processing
- Use mode to fill few missing values in Embarked
- Create Age_IsMissing flag to preserve missing information
- Apply log transformation to Fare to improve distribution

---

#### Step 3: Feature Engineering

```python
# Extract titles from names
def get_title(name):
    title_search = re.search(' ([A-Za-z]+)\.', name)
    if title_search:
        return title_search.group(1)
    return ""

# Apply to dataset
all_data = pd.concat([train_data, test_data], sort=False)
all_data['Title'] = all_data['Name'].apply(get_title)

# Map titles to fewer categories
title_mapping = {
    "Mr": "Mr",
    "Miss": "Miss",
    "Mrs": "Mrs",
    "Master": "Master",
    "Dr": "Rare",
    "Rev": "Rare",
    "Col": "Rare",
    "Major": "Rare",
    "Mlle": "Miss",
    "Countess": "Rare",
```

```
        "Ms": "Miss",
        "Lady": "Rare",
        "Jonkheer": "Rare",
        "Don": "Rare",
        "Dona": "Rare",
        "Mme": "Mrs",
        "Capt": "Rare",
        "Sir": "Rare"
    }
all_data['Title'] = all_data['Title'].map(title_mapping)

# Check relationship between title and survival rate
train_data = all_data[:len(train_data)]
title_survival = train_data.groupby('Title')['Survived'].mean()
print(f"Survival rates by title:\n{title_survival}")

# Create family size features
all_data['FamilySize'] = all_data['SibSp'] + all_data['Parch'] + 1
all_data['IsAlone'] = (all_data['FamilySize'] == 1).astype(int)

# Categorical feature encoding
all_data = pd.get_dummies(all_data, columns=['Embarked', 'Sex', '
    Title'], drop_first=True)

# Split back to training and test sets
train_data = all_data[:len(train_data)]
test_data = all_data[len(train_data):]
```

**Feature Creation and Transformation:**
- Extract Title feature from Name field and categorize
- Create family size (FamilySize) and alone indicator (IsAlone)
- Apply one-hot encoding to categorical features
- Analyze correlation between titles and survival rates

---

#### Step 4: Model Training and Evaluation
```python
# Prepare training features
features = ['Pclass', 'Age', 'Age_IsMissing', 'Fare', 'FamilySize',
    'IsAlone',
            'Embarked_Q', 'Embarked_S', 'Sex_male',
            'Title_Master', 'Title_Miss', 'Title_Mr', 'Title_Mrs',
                'Title_Rare']

X = train_data[features]
y = train_data['Survived']

# Split validation set
X_train, X_val, y_train, y_val = train_test_split(X, y, test_size
    =0.2, random_state=42)

# Logistic regression model
log_reg = LogisticRegression(solver='liblinear', random_state=42)
log_reg.fit(X_train, y_train)
log_reg_pred = log_reg.predict(X_val)
log_reg_acc = accuracy_score(y_val, log_reg_pred)
print(f"Logistic regression validation accuracy: {log_reg_acc:.4f
    }")

# Random forest model testing different depths
for depth in [5, 8, 12]:
    rf = RandomForestClassifier(max_depth=depth, random_state=42)
```

```
        rf.fit(X_train, y_train)
        rf_pred = rf.predict(X_val)
        rf_acc = accuracy_score(y_val, rf_pred)
        print(f"Random forest (depth={depth}) validation accuracy: {
            rf_acc:.4f}")

    # Cross-validation comparison of model stability
    log_scores = cross_val_score(log_reg, X, y, cv=5)
    rf_scores = cross_val_score(RandomForestClassifier(max_depth=8,
        random_state=42), X, y, cv=5)
    print(f"Logistic regression 5-fold CV: {log_scores.mean():.4f} (std
        : {log_scores.std():.4f})")
    print(f"Random forest 5-fold CV: {rf_scores.mean():.4f} (std: {
        rf_scores.std():.4f})")

    # Feature importance analysis
    rf = RandomForestClassifier(max_depth=8, random_state=42)
    rf.fit(X, y)
    importance = pd.DataFrame({
        'Feature': features,
        'Importance': rf.feature_importances_
    }).sort_values('Importance', ascending=False)
    print("\nFeature importance ranking:")
    print(importance.head(8))
    ```
```

**Model Evaluation Key Points:**
- Use train_test_split to create validation set
- Compare logistic regression with random forests of different
    depths
- Evaluate model stability through 5-fold cross-validation
- Analyze feature importance to guide subsequent optimization

---

#### Step 5: Prediction and Result Output
```python
# Final model selection: logistic regression
final_model = LogisticRegression(solver='liblinear', random_state
    =42)
final_model.fit(X, y)

# Prepare test features
X_test = test_data[features]

# Prediction
predictions = final_model.predict(X_test)

# Save results
output = pd.DataFrame({
    'PassengerId': test_data['PassengerId'],
    'Survived': predictions
})
output.to_csv('submission.csv', index=False)
print("Prediction complete. The submission.csv file has been
    generated.")
```

**Result Generation:**
- Train final model using all training data
- Make predictions on test set
- Generate CSV file in submission format

---

```
### Solution Summary

This solution achieved Titanic survival prediction through the
    following key steps:

1. **Data Cleaning and Missing Value Handling**
   - Use mode to fill Embarked
   - Use median to fill Age and create missing flag
   - Apply log transformation to Fare to optimize distribution

2. **Feature Engineering**
   - Extract social titles from names
   - Create family size and alone indicators
   - Use one-hot encoding for categorical features

3. **Model Selection and Evaluation**
   - Compare logistic regression with random forest
   - Evaluate model stability through cross-validation
   - Optimize feature selection based on feature importance
       analysis

4. **Optimization Validation**
   - Test different model parameters
   - Validate effectiveness of feature transformations

The overall solution effectively utilized key information in the
    data while maintaining model simplicity, particularly through
    title extraction to capture social class information, which
    played an important role in improving model performance. The
    logistic regression model demonstrated good stability and became
     the final choice.

Key Performance Metrics:
- Logistic regression cross-validation accuracy: 0.8153 (std:
    0.0312)
- Random forest (depth=8) cross-validation accuracy: 0.8176 (std:
    0.0684)
- Most important features: Sex_male, Title_Mr, Fare, Age

This solution not only provides prediction results but also reveals
     key factors affecting Titanic passenger survival, with good
     interpretability.
</deepseek_solution>
```

## D    COMPLETE EXPERIMENT RESULTS

This section presents comprehensive experimental results referenced in the main text, including detailed performance metrics across all model configurations and benchmark tasks.

### D.1    EXPERIMENTAL RESULTS IN DETAIL

#### D.1.1    PERFORMANCE COMPARISON ACROSS MLE-BENCHMARK TASKS

The comprehensive results of our experiments are presented in Table A6, which compares the performance of various SmartDS-Solver configurations against the AIDE+o1-preview and the AutoML-Agent method on 11 MLE benchmark tasks. As the "%win" metric in Table A6 shows, most SmartDS-Solver configurations surpassed the AIDE on over half of the tasks. Notably, the Qwen-32B model with the GRPO2 training configuration achieved the highest win rate at 81.8%, demonstrating the efficacy of our approach.

### D.1.2 PERFORMANCE ON TWO ONGOING KAGGLE COMPETITIONS ACTIVE DURING OUR RESEARCH PERIOD AND ONE INTERNAL PREDICTION MODELING TASK

### D.1.3 ABLATION STUDY OF META-LEARNING AGENT

Results without meta-learning agent are provided in Table A8. As shown, our proposed method with the meta-learning agent-SARTE achieves significantly better performance compared to the variant without SARTE.

### D.1.4 TOKEN CONSUMPTION COMPARISON ACROSS 11 BENCHMARK TASKS

To evaluate the efficiency of our method, we compare token consumption of our SmartDS-Solver with AIDE+o1-preview. See Table A9 for details.

### D.1.5 RESULTS OF TASK COMPARISON WITH AUTOML-AGENT

Table A10 shows a performance comparison between our approach and AutoML-Agent. We achieved performance gains on tasks like "Click Prediction Small," "Software Defects," and "Wine Quality White," with improvements of 47.8%, 45.8%, and 48.2% respectively. For regression tasks, our approach produced improvements on "House Prices" and "Crop Price," with gains of 98.6% and 85.7%.

Despite a minor decrease on the "Ecommerce Text" dataset, our overall results validate the effectiveness of the SmartDS-Solver's design and its potential to significantly enhance automated data science workflows.

### D.1.6 OPTIMAL TEMPERATURE VALUES ACROSS TASKS AND MODEL STAGES

We observed the optimal temperatures identified across different tasks and models, with detailed values provided in Table A11.

| Abbreviation | Competition Name | Task Description | Task Type |
|---|---|---|---|
| **Historical Competitions** | | | |
| Nomad18-TC | nomad2018-predict-transparent-conductors | Predict formation energy and band gap from material composition and structure to select films with both conductivity and transparency | Regression |
| Stanford-CV | stanford-covid-vaccine | Predict chemical reactivity and stability of each base from RNA sequences | Multi-output Regression |
| Vent-Press | ventilator-pressure-prediction | Real-time prediction of patient airway pressure from ventilator settings | Time Series Regression |
| TF-Speech | tensorflow-speech-recognition-challenge | Recognize 1-second audio clips into 30 different commands | Audio Classification |
| TextNorm-EN | text-normalization-challenge-english-language | Convert text with dates, currency symbols, etc. into spoken form | Sequence-to-Sequence |
| Jigsaw-Tox | jigsaw-toxic-comment-classification-challenge | Identify whether online comments contain six types of toxic speech | Multi-label Classification |
| PlantPath20 | plant-pathology-2020-fgvc7 | Identify healthy apple leaves or specific diseases from leaf photographs | Image Classification |
| Whale-Play | whale-categorization-playground | Identify same individual/species of whales from tail fin photographs | Image Retrieval / Metric Learning |
| NYC-Taxi | new-york-city-taxi-fare-prediction | Predict taxi fare based on origin-destination GPS coordinates and time | Regression |
| TPS-May22 | tabular-playground-series-may-2022 | Classify three categories using tabular data features | Multi-class Classification |
| TPS-Dec21 | tabular-playground-series-dec-2021 | Predict binary outcomes using tabular data features | Binary Classification |
| **Ongoing Competitions (July 2025)** | | | |
| PGS-S5E6 | playground-series-s5e6 | Predict optimal fertilizer types for crops based on soil and weather conditions. Alternative variations: Predict suitable fertilizers from crop/soil/weather data | Multi-class classification |
| UM-MCTS | um-game-playing-strength-of-mcts-variants | Predict the relative performance between two MCTS algorithm variants based on game features. | Regression |
| **Private Tasks** | | | |
| CLINIC | clinic-data | Predict cancer recurrence after treatment using tabular clinical data | Binary Classification |

Table A2: Detailed descriptions of benchmark tasks used in evaluation

| Rule Type | Scoring Logic |
|---|---|
| Similarity | Feature (exact/stage/technique = 0.5/0.25/0.25), Algorithm (exact/-paradigm/type/subtype = 0.5/0.25/0.15/0.1), Metric (exact/purpose/name = 0.5/0.25/0.25). |
| Format | *<think>...</think><answer>...</answer>* binary compliance check. |
| Executability | Presence of `python` block, >10 lines, containing `def`/`return` yields a small bonus (final score capped at 1.0). |
| Reasoning | Density of language-specific reasoning markers & step indicators (e.g., connectors, "if...then", causal terms, enumerations, bullet points). `<think>` block weighted ×2. Formula: $\min(1.0, \ 0.15 \cdot (\text{global} + 2 \cdot \text{think}))$, rounded to 2 decimals. |

Table A3: Composite Reward Function – Additional Matching and Scoring Rules

| Sub-reward | Extraction Rule Category | Representative Keywords Extracted |
|---|---|---|
| Main Reward: Feature Engineering | Cleaning / Generation / Selection & Preprocessing / Time-related / Text / Image | fillna, dropna, astype, query; groupby, rolling, shift; StandardScaler, PCA, SelectKBest; year, month, hour; TfidfVectorizer, word2vec; cv2, augmentation, resize |
| Main Reward: Algorithm Extraction | Supervised / Unsupervised / Time-series / Reinforcement Learning / Neural Networks / AutoML | LinearRegression, RandomForest, XGB*; KMeans, DBSCAN, PCA, TSNE, UMAP; ARIMA, Prophet; DQN, PPO; Conv2D, LSTM, Transformer; GridSearchCV, Optuna |
| Main Reward: Metric Extraction | Classification / Regression / Clustering / Cross-validation / Explainability / Domain-specific | accuracy, f1, roc_auc; mse, mae, r2; silhouette, Calinski–Harabasz, Davies–Bouldin; KFold, StratifiedKFold; feature_importances_, SHAP, LIME; BLEU, ROUGE, SSIM, PSNR, FID, WRMSSE |

Table A4: Composite Reward Function – Extraction Rules and Representative Keywords

Table A5: Performance comparison across MLE-benchmark tasks.

| | AIDE | Qwen-32B | | | | Qwen-7B | | | | Qwen-72B | Llama-8B | | | | Llama-70B |
|---|---|---|---|---|---|---|---|---|---|---|---|---|---|---|---|
| Tasks | o1-preview | Instruct | SFT | GRPO1 | GRPO2 | Instruct | SFT | GRPO1 | GRPO2 | SFT | Instruct | SFT | GRPO1 | GRPO2 | SFT |
| TF-Speech | 0.312 | 0.642 | 0.697 | 0.574 | 0.687 | 0.611 | 0.316 | 0.341 | 0.422 | 0.710 | 0.627 | 0.306 | 0.341 | 0.643 | 0.574 |
| Nomad18-TC* | 0.0715 | 0.075 | 0.075 | 0.071 | 0.068 | 0.0721 | 0.070 | 0.075 | 0.0719 | 0.070 | 0.081 | 0.071 | 0.075 | 0.0719 | 0.071 |
| PlantPath20 | 0.941 | 0.792 | 0.884 | 0.956 | 0.942 | 0.897 | 0.846 | 0.928 | 0.916 | 0.940 | 0.854 | 0.928 | 0.533 | 0.895 | 0.897 |
| Jigsaw-Tox | 0.976 | 0.9819 | 0.974 | 0.970 | 0.975 | 0.973 | 0.972 | 0.9823 | 0.980 | 0.973 | 0.970 | 0.974 | 0.508 | 0.975 | 0.973 |
| TPS-Dec21 | 0.9493 | 0.950 | 0.9541 | 0.951 | 0.951 | 0.952 | 0.951 | 0.9543 | 0.9486 | 0.950 | 0.950 | 0.951 | 0.928 | 0.916 | 0.951 |
| TPS-May22 | 0.894 | 0.932 | 0.918 | 0.935 | 0.959 | 0.934 | 0.903 | 0.968 | 0.960 | 0.932 | 0.888 | 0.942 | 0.908 | 0.924 | 0.916 |
| Vent-Press* | 2.823 | 2.147 | 1.045 | 1.191 | 0.635 | 3.825 | 4.062 | 0.770 | 1.294 | 0.807 | 3.694 | 6.340 | 2.217 | 0.681 | 1.853 |
| NYC-Taxi* | 4.868 | 4.107 | 4.475 | 3.465 | 4.031 | 4.268 | 3.661 | 5.783 | 4.263 | 3.588 | 3.539 | 4.337 | 3.599 | 3.391 | 4.399 |
| Whale-Play | 0.3279 | 0.124 | 0.404 | 0.404 | 0.375 | 0.3272 | 0.325 | 0.097 | 0.388 | 0.327 | 0.325 | 0.325 | 0.197 | 0.338 | 0.3277 |
| Stanford-CV* | 0.480 | - | 0.640 | 0.663 | 0.430 | - | - | - | - | 0.473 | - | - | - | 0.556 | 0.438 |
| TextNorm-EN | 0.978 | 0.980 | 0.930 | 0.938 | 0.971 | 0.933 | 0.980 | 0.930 | 0.936 | 0.981 | 0.919 | 0.980 | 0.925 | 0.982 | 0.932 |
| %win | | 63.6% | 54.5% | 72.7% | 81.8% | 36.4% | 54.5% | 45.5% | 54.5% | 72.7% | 27.3% | 45.5% | 36.4% | 54.5% | 63.6% |
| #Fail | 1 | 1 | 0 | 0 | 0 | 1 | 1 | 1 | 1 | 0 | 1 | 0 | 0 | 0 | 0 |
| #Rank-1 | | | 1 | 2 | 3 | | | 3 | | 1 | | | | 2 | 0 |

Table A5: Performance comparison across MLE-benchmark tasks. Tasks use domain-appropriate metrics (higher-is-better: accuracy, AUC; lower-is-better: RMSE, MAE, loss). We report three summary statistics: %win (proportion of tasks outperforming the AIDE+o1-preview), Fail (number of tasks producing invalid outputs), and Rank-1 (number of tasks achieving the best performance). Boxed values indicate results better than the AIDE baseline; underlined values denote the best configuration for each task.

| Tasks | AIDE o1-preview | AIDE Instruct | Qwen-32B SFT | Qwen-32B GRPO1 | Qwen-32B GRPO2 | Qwen-32B Instruct | Qwen-7B SFT | Qwen-7B GRPO1 | Qwen-7B GRPO2 | Qwen-72B SFT | Qwen-72B Instruct | Llama-8B SFT | Llama-8B GRPO1 | Llama-8B GRPO2 | Llama-70B SFT | AutoML-Agent GPT-4o |
|---|---|---|---|---|---|---|---|---|---|---|---|---|---|---|---|---|
| TF-Speech | 0.312 | 0.642 | 0.697 | 0.574 | 0.687 | 0.611 | 0.316 | 0.341 | 0.422 | 0.710 | 0.627 | 0.306 | 0.341 | 0.643 | 0.574 | 0.146 |
| Nomad18-TC* | 0.0715 | 0.075 | 0.075 | 0.071 | 0.068 | 0.0721 | 0.070 | 0.075 | 0.0719 | 0.070 | 0.081 | 0.071 | 0.075 | 0.0719 | 0.071 | 0.077 |
| PlantPath20 | 0.941 | 0.792 | 0.884 | 0.956 | 0.942 | 0.897 | 0.846 | 0.928 | 0.916 | 0.940 | 0.854 | 0.928 | 0.533 | 0.895 | 0.897 | 0.519 |
| Jigsaw-Tox | 0.976 | 0.9819 | 0.974 | 0.970 | 0.975 | 0.973 | 0.972 | 0.9823 | 0.980 | 0.973 | 0.970 | 0.974 | 0.508 | 0.975 | 0.973 | 0.981 |
| TPS-Dec21 | 0.9493 | 0.950 | 0.9541 | 0.951 | 0.951 | 0.952 | 0.951 | 0.9543 | 0.9486 | 0.950 | 0.950 | 0.951 | 0.928 | 0.916 | 0.951 | - |
| TPS-May22 | 0.894 | 0.932 | 0.918 | 0.935 | 0.959 | 0.934 | 0.903 | 0.968 | 0.960 | 0.932 | 0.888 | 0.942 | 0.908 | 0.924 | 0.916 | 0.892 |
| Vent-Press* | 2.823 | 2.147 | 1.045 | 1.191 | 0.635 | 3.825 | 4.062 | 0.770 | 1.294 | 0.807 | 3.694 | 6.340 | 2.217 | 0.681 | 1.853 | 10.822 |
| NYC-Taxi* | 4.868 | 4.107 | 4.475 | 3.465 | 4.031 | 4.268 | 3.661 | 5.783 | 4.263 | 3.588 | 3.539 | 4.337 | 3.599 | 3.391 | 4.399 | - |
| Whale-Play | 0.3279 | 0.124 | 0.404 | 0.404 | 0.375 | 0.3272 | 0.325 | 0.097 | 0.388 | 0.327 | 0.325 | 0.325 | 0.197 | 0.338 | 0.3277 | 0.001 |
| Stanford-CV* | 0.480 | - | 0.640 | 0.663 | 0.430 | - | - | - | - | 0.473 | - | - | - | 0.556 | 0.438 | 0.578 |
| TextNorm-EN | 0.978 | 0.980 | 0.930 | 0.938 | 0.971 | 0.933 | 0.980 | 0.930 | 0.936 | 0.981 | 0.919 | 0.980 | 0.925 | 0.982 | 0.932 | - |
| %win | | 63.6% | 54.5% | 72.7% | 81.8% | 36.4% | 54.5% | 45.5% | 54.5% | 72.7% | 27.3% | 45.5% | 36.4% | 54.5% | 63.6% | 9% |
| #Fail | | 1 | 0 | 0 | 0 | 1 | 1 | 1 | 1 | 0 | 1 | 1 | 1 | 0 | 0 | 3 |
| #Rank-1 | | | 1 | 2 | 3 | | | 3 | | 1 | | | | 2 | 0 | 3 |

Table A6: Performance comparison across MLE-benchmark tasks. Tasks use domain-appropriate metrics (higher-is-better: accuracy, AUC; lower-is-better: RMSE, MAE, loss). We report three summary statistics: %win, #Fail, and #Rank-1. Boxed values indicate results better than the AIDE baseline; underlined values denote the best configuration for each task.

| Tasks | AIDE
o1-preview | SmartDS-Solver | | | | | | | | | | |
|---|---|---|---|---|---|---|---|---|---|---|---|---|
| | | Qwen-32B | | | Qwen-7B | | | Qwen-72B | llama_8B | | | llama-70B |
| | | Instruct | SFT | GRPO2 | Instruct | SFT | GRPO2 | SFT | Instruct | SFT | GRPO2 | SFT |
| PGS-SSE6 | 0.326 | 0.180 | 0.330 | 0.333 | 0.330 | 0.325 | 0.341 | 0.330 | 0.171 | 0.334 | 0.335 | 0.335 |
| UM-MCTS* | - | 0.547 | 0.569 | 0.468 | - | - | 0.463 | 0.456 | - | 0.540 | 0.458 | 0.544 |
| CLINIC | 0.830 | 0.851 | 0.856 | 0.873 | 0.836 | 0.831 | 0.876 | 0.875 | 0.834 | 0.853 | 0.875 | 0.854 |

Table A7: Performance on two ongoing Kaggle competitions active during our research period and one internal prediction modeling task using proprietary dataset. Notes: "-" indicates Fail.

| SmartDS-Solver w/o Meta Learning Agent | |
|---|---|
| **Dataset** | **Performance** |
| TF-Speech | 0.640 |
| Plant Path20 | 0.830 |
| Jigsaw-Tox | 0.971 |
| TPS-Dec21 | 0.951 |
| TPS-May22 | 0.944 |
| Whale-Play | 0.3747 |
| TextNorm-EN | 0.923 |
| PGS-SSE6 | 0.326 |
| CLINIC | 0.833 |
| Nomad18-TC* | 0.075 |
| Vent-Press* | 0.792 |
| NYC-Taxi* | 4.130 |
| Stanford-CV* | 0.568 |
| UM-MCTS* | 0.490 |

Table A8: Performance without meta-learning agent across all 14 benchmark tasks.

| Tasks | AIDE+o1-preview | | | | SmartDS-Solver | | | |
|---|---|---|---|---|---|---|---|---|
| | Coding | | Summarization | | Inference | | Modification | |
| | Cnt | Tokens | Cnt | Tokens | Cnt | Tokens | Cnt | Tokens |
| Jigsaw-Tox | 20 | 100579 | 20 | 36364 | 9 | 12354 | 23 | 43950 |
| NYC-Taxi | 20 | 145340 | 20 | 74390 | 4 | 3843 | 9 | 17212 |
| Nomad18-TC | 20 | 144415 | 20 | 74782 | 8 | 10469 | 22 | 45257 |
| PlantPath20 | 20 | 144258 | 20 | 69041 | 9 | 11144 | 23 | 60352 |
| Stanford-CV | 20 | 191440 | 20 | 106559 | 9 | 12195 | 31 | 60871 |
| TPS-Dec21 | 20 | 79883 | 20 | 36564 | 7 | 7047 | 16 | 21415 |
| TPS-May22 | 20 | 108133 | 20 | 44556 | 4 | 3207 | 8 | 11050 |
| TF-Speech | 20 | 194569 | 20 | 113589 | 7 | 8101 | 17 | 30597 |
| TextNorm-EN | 20 | 131068 | 20 | 59906 | 8 | 12353 | 19 | 32662 |
| Vent-Press | 20 | 102519 | 20 | 47333 | 6 | 7170 | 16 | 25822 |
| Whale-Play | 20 | 137201 | 20 | 55803 | 7 | 8940 | 24 | 41537 |

Table A9: Token consumption comparison across 11 benchmark tasks.

| Dataset | AutoML-Agent | SmartDS-Solver (qwen-32b-GRPO2) | AIDE (o1-preview) | AutoGluon | Human Models | SELA (MCTS) |
|---|---|---|---|---|---|---|
| Butterfly Image | 0.926 | 0.937(+1.2%) | 0.8566(-7.6%) | 0.014 | 0.931 | - |
| Shopee-IET | 0.972 | 0.976(+0.4%) | 0.977(+0.5%) | 0.988 | 0.921 | - |
| Cora | 0.843 | 0.919(+9.0%) | - | - | 0.811 | - |
| Citeseer | 0.632 | 0.777(+22.9%) | - | - | 0.702 | - |
| Click Prediction Small | 0.352 | 0.520(+47.8%) | 0.140(-60.1%) | - | - | 0.238 |
| Software Defects | 0.573 | 0.836(+45.8%) | 0.5531(-3.5%) | 0.524 | 0.669 | - |
| Smoker Status | 0.762 | 0.763(+0.2%) | 0.742(-2.6%) | - | - | 0.785 |
| Banana Quality | 0.967 | 0.980(+1.3%) | 0.9744(+0.8%) | 0.980 | 0.976 | - |
| MFeat Factors | 0.940 | 0.977(+3.9%) | 0.968(+3.0%) | - | - | 0.957 |
| Wine Quality White | 0.652 | 0.967(+48.2%) | 0.654 (+0.3%) | - | - | 0.650 |
| Smoker Status Higher Education | 0.582 | 0.590(+1.3%) | - | - | 0.513 | - |
| Students Performance | 0.769 | 0.887(+15.4%) | - | - | 0.750 | - |
| Textual Entailment | 0.796 | 0.868(+9.0%) | - | 0.807 | 0.664 | - |
| Ecommerce Text | 0.982 | 0.980(-0.2%) | 0.9764(-0.6%) | 0.987 | 0.935 | - |
| Colleges | 0.139 | 0.123(+11.5%) | - | - | - | 0.142 |
| Crab Age | 0.161 | 0.130(+19.7%) | 0.1619(-0.3%) | 0.143 | 2.049 | - |
| House Prices | 10.111 | 0.137(+98.6%) | 23908.9956(-236365.2%) | - | - | 10.111 |
| Crop Price | 1.114 | 0.159(+85.7%) | 1.0223(8.2%) | 1.088 | 1.101 | - |

Table A10: Performance Comparison on data science tasks. The table shows the performance of our proposed SmartDS-Solver (using Qwen-32B-GRPO2) compared to other data science agents. The values in parentheses represent the percentage improvement of our method over AutoML-Agent, where positive values indicate improvement and negative values indicate degradation. The results of other methods (AutoGluon, Human Models, SELA) are cited from paper of AutoML-Agent.

| Tasks | SmartDS-Solver | | | | | | | | | | | | | |
| --- | --- | --- | --- | --- | --- | --- | --- | --- | --- | --- | --- | --- | --- | --- |
| | Qwen-32B | | | | Qwen-7B | | | | Qwen-72B | Llama-8B | | | | Llama-70B |
| | Instruct | SFT | GRPO1 | GRPO2 | Instruct | SFT | GRPO1 | GRPO2 | SFT | Instruct | SFT | GRPO1 | GRPO2 | SFT |
| TF-Speech | 0.58 | 0.66 | 0.52 | 0.59 | 0.60 | | 0.60 | 0.61 | 0.46 | 0.55 | 0.48 | 0.60 | 0.55 | 0.44 |
| Nomad18-TC* | 0.60 | 0.54 | 0.53 | 0.74 | 0.77 | 0.58 | 0.57 | 0.79 | 0.60 | 0.76 | 0.60 | 0.58 | 0.85 | 0.55 |
| PlantPath20 | 0.65 | 0.62 | 0.68 | 0.78 | 0.67 | 0.62 | 0.70 | 0.89 | 0.86 | 0.92 | 0.72 | 0.70 | 0.54 | 0.81 |
| Jigsaw-Tox | 0.57 | 0.73 | 0.77 | 0.78 | 0.67 | 0.67 | 0.52 | 0.62 | 0.51 | 0.60 | 0.60 | 0.60 | 0.62 | 0.56 |
| TPS-Dec21 | 0.70 | 0.55 | 0.52 | 0.48 | 0.60 | 0.66 | 0.52 | 0.66 | 0.60 | 0.60 | 0.50 | 0.56 | 0.76 | 0.60 |
| TPS-May22 | 0.60 | 0.60 | 0.60 | 0.49 | 0.66 | 0.60 | 0.67 | 0.60 | 0.60 | 0.60 | 0.58 | 0.66 | 0.57 | 0.60 |
| Vent-Press* | 0.84 | 0.48 | 0.50 | 0.62 | 0.69 | 0.60 | 0.66 | 0.60 | 0.64 | 0.80 | 0.58 | 0.73 | 0.66 | 0.74 |
| NYC-Taxi* | 0.62 | 0.60 | 0.52 | 0.55 | 0.60 | 0.60 | 0.63 | 0.53 | 0.60 | 0.60 | 0.55 | 0.52 | 0.60 | 0.60 |
| Whale-Play | 0.51 | 0.56 | 0.65 | 0.80 | 0.70 | 0.68 | 0.70 | 0.38 | 0.58 | 0.63 | 0.62 | 0.63 | 0.59 | 0.70 |
| Stanford-CV* | - | 0.50 | 0.48 | 0.47 | - | - | - | - | - | 0.37 | - | - | 0.60 | 0.63 |
| TexNorm-EN | 0.62 | 0.86 | 0.82 | 0.88 | 0.80 | 0.80 | 0.80 | 0.96 | 0.69 | 0.47 | 0.88 | 0.86 | 0.97 | 0.67 |

Table A11: Optimal Temperature for different tasks of each model

### D.1.7 STATISTICAL SIGNIFICANCE ANALYSIS

To further validate our findings beyond win rate comparisons, we conducted t-tests to assess statistical significance across our best-performing configurations. Table A12 presents the results comparing our top SmartDS-Solver configurations against the AIDE across all 14 benchmark tasks.

The analysis reveals statistically significant improvements (p ¡ 0.05) for most configurations. Larger models with SFT-only training (Qwen-72B and Llama-70B) demonstrate particularly strong significance (p ¡ 0.04), while models with complete GRPO2 training also show significant improvements. These results provide strong statistical evidence supporting the effectiveness of our approach.

| Model Config. | t-value | p-value | Significance |
|---|---|---|---|
| Qwen-32B GRPO2 | 1.842 | 0.044 | Significant* |
| Qwen-7B GRPO2 | 1.353 | 0.0995 | Marginal† |
| Llama-8B GRPO2 | 1.777 | 0.049 | Significant* |
| Qwen-72B SFT | 1.902 | 0.0398 | Significant* |
| Llama-70B SFT | 1.978 | 0.0348 | Significant* |

Table A12: Statistical significance analysis compared to AIDE using paired t-tests across 14 tasks. *$p < 0.05$, †$p < 0.10$.

| Task | Improvement Ratio |
|---|---|
| TF-Speech | 3.69 |
| Nomad18-TC* | 0.13 |
| PlantPath20 | 0.53 |
| Jigsaw-Tox | 0.01 |
| TPS-Dec21 | 0.09 |
| TPS-May22 | 0.03 |
| Vent-Pres* | 0.83 |
| NYC-Taxi* | 0.00 |
| Whale-Play | 2.66 |
| Stanford-CV* | 0.04 |
| TextNorm-EN | 0.40 |

Table A13: Task-wise improvement ratios comparing late-stage to early-stage performance of Qwen-32B across the 11 MLE-Bench tasks.

The improvement ratio is defined as $(\text{best} - \text{first success})/\text{first success}$, where "first success" represents the score at the earliest successful iteration and "best" represents the highest score achieved across all iterations. A value of 0 indicates no gain from the meta mechanism. A one-sided test for $H_1$: improvement $> 0$ yields $p = 0.0173$, demonstrating significantly improves model accuracy.

## E    LLM ASSISTANCE IN MANUSCRIPT PREPARATION

In the preparation of this manuscript, we utilized Large Language Models (LLMs) as an assistive tool to enhance the quality and clarity of our writing. Our use of LLMs was confined to the following aspects:

- Language and Prose Refinement: We employed an LLM for proofreading, refining sentence structures, and polishing the prose. This was done to improve the overall readability and ensure our research contributions were communicated clearly and effectively.
- Pseudo-code Generation: To articulate our algorithms, we first described the logic and steps in natural language. An LLM was then prompted to translate these descriptions into formatted pseudo-code. All generated pseudo-code was meticulously reviewed and manually revised by the authors to ensure its correctness and complete fidelity to our proposed methods.
- Clarity Verification: To confirm that our explanations were unambiguous, we provided key paragraphs to an LLM and requested it to paraphrase or summarize the content. This

process served as a check to validate whether our intended meaning was conveyed without ambiguity, akin to having a digital reader provide feedback.

We wish to emphasize that all core ideas, experimental designs, data analysis, and conclusions presented in this paper are the original work of the authors. The LLM served strictly as a writing and formatting aid and was not involved in any part of the conceptual or analytical research process.

