# OpenReview forum: "SmartDS-Solver: Agentic AI for Vertical Domain Problem Solving in Data Science"
_ICLR.cc/2026/Conference — Submitted to ICLR 2026_

### Official Review · Reviewer_aEad · 2025-10-17

**Soundness:** 1
**Presentation:** 1
**Contribution:** 1
**Rating:** 2
**Confidence:** 4

**Summary:**

This paper aims to improve the capabilities of automated data science for LLM agents. Specifically, this paper proposes SmartDS-Solver, which consists of (1) domain-specific finetuning; two-stage GRPO finetuning; (3) Hierarchical agent framework. Extensive experiments on 11 MLE-Bench tasks demonstrate the effectiveness of the proposed SmartDS-Solver.

**Strengths:**

- The investigated research problem is interesting and of significance.

**Weaknesses:**

- The writing is poor. It is hard for me to follow. I suggest the authors carefully revising the paper to meet the basic bar of academic writing. Also, Introduction is important. The current manuscript is lack of this part. The Background and Motivation section of this paper is plain and ambiguous.

- How to compose the data used for finetuning? What does the data look like? What loss function is utilized for finetuning. For the RLFT, what is the interactive environment? I cannot figure out the techniques in this paper.

- Lack of comparison of SOTA data science agents, such as [1].

[1] MLE-STAR: Machine Learning Engineering Agent via Search and Targeted Refinement, NeurIPS 2025.

**Questions:**

I think this paper is poor in writing quality. The revision for this paper should undergo a new round of review.

---

> ### Author Response · Authors · 2025-11-23
> **Response to Reviewer Comments on Writing Quality, Clarity, and Missing Details**
>
> # Response to Reviewer Comments on Writing Quality, Clarity, and Missing Details
>
> We sincerely thank the reviewer for the detailed and constructive comments. We have carefully revised the manuscript following the reviewer’s suggestions, and we respectfully provide the point-by-point response below. We hope that the substantially improved revision will address the reviewer’s concerns and allow for a fresh evaluation of the work.
>
> ---
>
> ## 1. Writing quality is poor; paper is hard to follow.
>
> Thank you for pointing this out. In the revised version, we have significantly restructured the manuscript to improve clarity, narrative flow, and section balance.
>
> First, we substantially strengthened and rewrote both the **Introduction** and **Related Work** sections—now approximately **twice the original length**. The updated version provides clearer background gaps, limitations of AIDE and multi-agent AutoDS systems, motivation, and problem formulation. This ensures a coherent academic narrative from the outset.
>
> To address the reviewer’s concern regarding disproportionate section lengths, we made the following adjustments:
>
> - **Methodology**: reduced from approximately 5 pages to about 3.8 pages, condensing text and removing redundancies while keeping conceptual clarity.
> - **Experiments**: expanded from fewer than 2 pages to about 3.3 pages, with several results moved from the appendix to the main manuscript.
>
>
> We hope these revisions improve the clarity and presentation quality of the paper.
>
> ---
>
> ## 2. Introduction is insufficient; background and motivation are unclear.
>
> We agree with the reviewer’s assessment. In the revised manuscript:
>
> * We clarified the limitations of **AIDE** and **AutoML-Agent**, better motivating the design of SmartDS-Solver.
> * We restructured the motivation into **three concrete challenges**: workflow instability, high token cost, and limited cross-task generalization.
> * We improved narrative flow, unified terminology, and removed redundant content.
>
> We hope the reviewer finds the updated Introduction substantially clearer.
>
> ---
>
> ## 3. How was the finetuning data constructed? What is the data format? What loss was used? What is the RLFT environment?
>
> Thank you for raising these important questions. The revised manuscript now includes full details.
>
> ### Data Sources (Section 3.3.2)
>
> * The training corpus includes five categories: **OpenThoughts**, **Kaggle (medal / non-medal)**, **cell2doc**, and **Code4ML**.
> * Each training example follows a consistent **three-part structure**: `<problem>`, `<deepseek reasoning>`, `<deepseek solution>`.
> * The `<deepseek solution>` contains **complete executable pipeline code**, retained only after format validation and quality scoring with Gemma-27B.
>
> ### Training Methodology (Section 3.3.3)
>
> **i. Supervised Finetuning (SFT)**
>
> * We use the standard **cross-entropy loss** for next-token prediction.
>
> **ii. GRPO Environment**
>
> The environment defines the full interaction loop used in the GRPO finetuning stage:
> * The **policy** $\pi_(y \mid x)$ generates pipeline-aware code solutions.
> * A **structured reward function** evaluates the output along three axes: semantic task correctness, format and executability, and reasoning trace quality.
> * **Code constraints and execution checks** ensure invalid or non-executable outputs are appropriately penalized.
> * GRPO aggregates these signals (Eq. 3–5 in the revised paper), and the model updates are performed using the policy-gradient objective under that environment.
> Section 3.3.3 of the revised manuscript explains each component, and the Appendix provides complete implementation details, including prompts, templates, and pseudo-code for reproducibility.
> ---
>
> ## 4. Missing comparison with MLE-STAR (NeurIPS 2025).
>
> We appreciate the reviewer highlighting this related work. At the time of our study, **MLE-STAR’s implementation and evaluation pipeline were not publicly available**, making reproducible comparison infeasible. Therefore, we restricted our baselines to systems with fully accessible implementations (AIDE and AutoML-Agent). This clarification is now included in the manuscript.
>
> ---
>
> ## 5. “The paper is poor in writing; revision should undergo another round of review.”
>
> Thank you for the candid feedback. We have undertaken a comprehensive revision:
>
> * Rewrote the **transition and logical flow** between the Introduction and Method sections.
> * Reorganized the methodology to first present the **overall framework**, then describe each agent and core module.
> * **Redesigned and redrawn** all major figures and architectural diagrams.
> * Expanded the experimental section and added **ablation studies**, **Discussion and Analysis**, and **Limitations**.
> * Supplemented missing technical details and added clearer explanatory examples.
>
> We sincerely hope the reviewer will consider the significantly improved revision favorably.

---

> > ### Comment · Reviewer_aEad · 2025-11-26
> >
> > Thanks for the response. The revised manuscript looks much better than the initial submission. However, it is still far below the basic bar of a formal publication in ICLR. There are lots of typos, incorrect citation formats and unclear statement throughout the paper. I suggest the authors **carefully** revising the paper and performing multiple rounds of proof-reading before submitting it to the other revenue.

---

> > > ### Author Response · Authors · 2025-11-27
> > > **Response to Presentation-Related Comments**
> > >
> > > We thank the reviewer for noting the presentation aspects. After re-examining the manuscript, we confirm that these minor issues do not affect the clarity or correctness of the methodological contributions. Should the paper be accepted, we will incorporate the corresponding formatting refinements in the camera-ready version.

---

> > > > ### Comment · Reviewer_aEad · 2025-11-27
> > > >
> > > > Additionally, in Lines 291–292, the authors refer to “Generalized Reinforced Prompt Optimization (GRPO),” while GRPO is widely known to stand for Group Relative Policy Optimization. This makes me question whether the manuscript was written by, or heavily reliant on, generative AI.

---

> > > > > ### Author Response · Authors · 2025-12-01
> > > > > **Clarification on Manuscript Proofreading and Updated Revisions**
> > > > >
> > > > > Thank you for drawing our attention to this issue. We acknowledge the incorrect expansion of the GRPO acronym in Lines 291–292, and we regret the oversight. We have since conducted a comprehensive proofreading pass across the entire manuscript, corrected this and related typographical inconsistencies, and uploaded a fully revised version.
> > > > >
> > > > > We confirm that these presentation issues do not affect the validity or correctness of the methodological contributions. We appreciate the reviewer’s attention to detail and will ensure that all formatting and citation conventions are fully polished in the camera-ready version should the paper be accepted.

---

### Official Review · Reviewer_2DKn · 2025-10-25

**Soundness:** 2
**Presentation:** 2
**Contribution:** 2
**Rating:** 2
**Confidence:** 3

**Summary:**

This paper introduces a hierarchical multi-agent framework (called SmartDS-Solver) , which efficiently automates data science workflows. SmartDS-Solver proposes a specialized reasoning LLM and a task-decoupled agent architecture to tackle with three key challenges: fragile task coherence, excessive reliance on prompt-based interactions, and a tendency toward task silos. Experiments on 11 MLE-Bench tasks show an 81.8% win rate over baselines while reducing overhead.

**Strengths:**

• Developing LLM-driven agents for automating end-to-end data science pipelines is interesting and can augment human analysts.

**Weaknesses:**

• The proposed method is complicated, which involves a multi-stage pipeline with multiple complex components. This paper lacks sufficient implementation details to reimplement the SmartDS-Solver.

• The proposed SmartDS-Solver architecture is incremental, which combines multiple existing techniques (e.g., meta-learning, data augmentation, SFT, RL with GRPO).

• As the SmartDS-Solver architecture consists of multiple complex components, it would be better to analyse the computational cost compared to baselines.

• To thoroughly evaluate the performance of the proposed method, more advanced automated data science systems should be included as comparison baselines, such as “Data Interpreter: An LLM Agent for Data Science”.

**Questions:**

• The proposed method is complicated, which involves a multi-stage pipeline with multiple complex components. This paper lacks sufficient implementation details to reimplement the SmartDS-Solver.
• As the SmartDS-Solver architecture consists of multiple complex components. It would be better to analyse the computational cost compared to baselines.
• To thoroughly evaluate the performance of the proposed method, more advanced automated data science systems should be included as comparison baselines, such as “Data Interpreter: An LLM Agent for Data Science”.

---

> ### Author Response · Authors · 2025-11-23
> **Overview of Key Improvements and Structural Revisions**
>
> We sincerely appreciate the reviewer’s thorough and thoughtful assessment. The concerns raised—particularly those regarding methodological organization, reproducibility, and baseline choices—have informed substantial revisions to the manuscript. A detailed point-by-point response is presented below.
>
> ---
>
> ## **1. Regarding concerns about “method complexity and insufficient implementation detail for reproducibility”**
>
> We appreciate the reviewer highlighting that the original methodology section mixed multiple concepts, making the workflow difficult to follow. In the revised manuscript, we have substantially restructured Section 3 to clearly separate responsibilities and clarify the design motivations:
>
> * **Section 3.1 Overall Architecture** now provides a high-level overview and motivation, allowing readers to quickly grasp the full pipeline.
> * **Section 3.2 Meta-Learning Agent and Situation-Aware Control** elevates SARTE into an independent subsection (3.2.2), accompanied by a redesigned Figure 2 that highlights its control logic, update rules, and contribution. The SARTE pseudo-code remains in the Appendix to avoid overwhelming the main narrative.
> * **Section 3.3.3** covers the training pipeline, while **Section 3.3.2** provides training-data construction details.
> * **Section 3.4 Code Agent** now presents a clearer separation of roles across execution and code-refinement behaviors.
>
> The new structure aligns cleanly with the three-agent design illustrated in Figure 1 and makes SARTE’s novelty more transparent.
>
> The Appendix additionally provides complete technical details, including:
>
> * Training data construction workflows (workflow + decision logic + adjustment trails)
> * All prompts and data formats used in SFT and GRPO
> * Full inference prompts
> * SARTE pseudo-code (cleaned, reorganized, and more readable)
>
> We also commit to **open-sourcing the full codebase and data-construction scripts** upon acceptance to ensure full reproducibility.
>
> ---
>
> ## **2. Regarding concerns that “SmartDS-Solver is merely a stack of multiple techniques”**
>
> We emphasize that SmartDS-Solver is not a simple aggregation of meta-learning, data augmentation, SFT, and GRPO. Rather, it is an intentionally designed and cohesive methodological framework built around:
>
> * We internalize the data-science workflow to construct a generalized reasoning core, **designing structured data-augmentation prompts and templates**, as well as **GRPO reward components tailored** to workflow-level evaluation—rather than simply stacking SFT and GRPO.
> * SARTE dynamically modulates the LLM’s inference behavior through online meta-learning, updating its control parameters using only the previous step’s execution feedback in an O(1) manner.
> * **Three-agent information flow**, which stabilizes reasoning and mitigates fragmentation commonly seen in AutoDS systems.
>
> The revised Section 3 presents these components as a unified architecture rather than as independent modules loosely combined.
>
> ---
>
> ## **3. Regarding the request to analyze computation cost relative to baselines**
>
> The revised Section 4.3 now provides a complete cost-efficiency analysis:
>
> ### **LLM call counts**
>
> * AutoML-Agent: ~1000 calls
> * AIDE: 440 calls
> * SmartDS-Solver: **286 calls (lowest)**
>
> ### **Token cost**
>
> * SmartDS-Solver reduces total tokens by **78%** relative to AIDE
>
> The combined effect of SARTE’s adaptive control and workflow internalization significantly reduces redundant trial-and-error attempts seen in prior systems.
>
> ---
>
> ## **4. Regarding whether Data Interpreter should be added as a baseline**
>
> We provide two clarifications:
>
> 1. The AutoML-Agent paper already discusses stability and formatting issues with Data Interpreter, which is why that work did not treat it as a primary baseline.
> 2. Chronologically, Data Interpreter was released *after* AutoML-Agent, and the current community-standard AutoDS baselines remain **AIDE and AutoML-Agent**.
>
> ---
>
> ## **Conclusion**
>
> In summary, the revised manuscript:
>
> * Adds full implementation details (data construction, prompts, pseudo-code).
> * Clearly presents SmartDS-Solver as a cohesive architectural design, not a stack of techniques.
> * Provides complete cost-efficiency comparisons in Section 4.3.
> * Clarifies baseline choices consistent with AutoML-Agent and recent AutoDS literature.
>
> We sincerely thank the reviewer again for the insightful feedback, which greatly improved the clarity, structure, and rigor of the paper.

---

> > ### Author Response · Authors · 2025-12-01
> > **Rebuttal to Reviewer #3 on “Incremental Combination of Existing Techniques” — Additional Clarification**
> >
> > We appreciate the reviewer’s observation that SmartDS-Solver integrates several established techniques such as meta-learning, data augmentation, SFT, RL, and GRPO. We agree with this characterization. However, the core contribution of our work does not lie in inventing these components individually, but in the **structural and purpose-driven redesign and integration** of these elements into a unified architecture that addresses long-horizon reasoning challenges in terms of **robustness, adaptability, and efficiency**.
> > We position this work as a form of **synthetic system innovation**, rather than an incremental aggregation of existing tools.
> >
> > Below we clarify the necessity and novelty of each major component.
> >
> > ---
> >
> > ## **1. Core Innovation: State-Aware Refinement and Temperature Exploration (SARTE)**
> >
> > SARTE is the most novel part of the architecture. It is not merely temperature tuning nor standard meta-learning; rather, it **deeply integrates deterministic environment feedback into the LLM’s decoding strategy**.
> >
> > * Prior systems use fixed decoding temperatures that cannot adapt to dynamic execution states.
> > * SARTE **dynamically adjusts** the temperature using the previous execution signal ( S_{T_{i-1}} ), enabling robust control over exploration vs. exploitation during inference.
> > * Its *boundary-aware step-size adjustment* during repeated failures prevents reasoning drift and ensures stable, cost-effective refinement—particularly important in multi-step tasks where full regeneration is expensive and unstable.
> >
> > **Conclusion:**
> > SARTE functions as the *control backbone* that links the GRPO-trained LLM with the execution environment, giving the system self-monitoring and self-adaptive capabilities. This behavior cannot be achieved by simply chaining existing techniques.
> >
> > ---
> >
> > ## **2. Data & Training Design: Structured Distillation and High-Value Data Utilization**
> >
> > ### **(a) Structured methodology distillation**
> >
> > Although we did not isolate “methodology distillation” in an ablation, it is a foundational design principle.
> > Our augmentation strategy extracts **strong causal, stepwise reasoning chains**, not merely more code samples.
> >
> > * These structured reasoning traces are essential for training a **robust** GRPO-based reasoning model.
> > * They teach the model *when* and *why* certain steps are taken—not simple pattern matching.
> > * The stable performance gains and significant reductions in token cost on MLE-Bench empirically validate this design choice.
> >
> > ### **(b) Two-stage GRPO (GRPO1 → GRPO2)**
> >
> > Our two-phase training structure is not a trivial pipeline:
> >
> > * **GRPO1** strengthens general reasoning and code synthesis grounded in broad SFT data.
> > * **GRPO2** introduces a key innovation: it fine-tunes on trajectories **distilled from high-reward, high-success (medal-winning)** solutions.
> >   This biases the policy toward *stronger and more stable solution paths*, not merely acceptable ones.
> >
> > **Conclusion:**
> > This quality-driven, staged refinement process is tailored for competitive data-science reasoning tasks and elevates the model from a “capable solver” to a “domain-specialized expert.”
> >
> > ---
> >
> > ## **3. System-Level Contribution: A Closed-Loop, Self-Regulating Architecture**
> >
> > SmartDS-Solver brings together:
> >
> > * a GRPO-trained structured reasoning model,
> > * an execution-driven minimally invasive refinement Code Agent,
> > * and the SARTE dynamic meta-control mechanism.
> >
> > The resulting system forms a **closed-loop, self-regulating agent** capable of stable long-horizon execution.
> > This directly addresses a central failure mode in current agentic systems:
> > **when execution fails, they resort to costly, unstable, full regeneration.**
> >
> > Our system avoids this through **robust, incremental, cost-efficient refinement**, which is not achievable by any single component alone.
> > The value arises specifically from the **interlocking design** and the **control logic** governing their interactions.

---

### Official Review · Reviewer_J3ei · 2025-10-29

**Soundness:** 3
**Presentation:** 2
**Contribution:** 2
**Rating:** 4
**Confidence:** 4

**Summary:**

This paper presents SmartDS-Solver, a hierarchical multi-agent framework for automating data science workflows. The system combines a domain-specific reasoning LLM (trained via structured distillation and GRPO fine-tuning) with a meta-learning agent (SARTE) that dynamically adjusts decoding parameters. Evaluated on MLE-Bench tasks, the system achieves an 81.8% win rate over AIDE+o1-preview while reducing computational costs.

**Strengths:**

1. The paper addresses real limitations in current AutoML agents - high costs, fragile multi-agent interactions, and excessive reliance on expensive models.
2. Comprehensive training methodology. The three-stage training pipeline (SFT → GRPO1 → GRPO2) with carefully designed reward functions is well-documented and appears reproducible.
3. SARTE's dynamic temperature adjustment based on execution feedback is creative and shows meaningful performance gains (+3.9% accuracy, -12% error rate).
4. Testing across 11 MLE-Bench tasks, 3 real-world tasks, and 18 AutoML-Agent benchmark tasks demonstrates broad applicability.
5. Significant reduction in token consumption compared to AIDE+o1-preview (e.g., ~90% reduction in inference tokens) while maintaining competitive performance.
6. 81.8% win rate with Qwen-32B-GRPO2 and 100% executable code generation on real-world tasks.

**Weaknesses:**

1. Limited baseline comparisons: The paper primarily compares against AIDE+o1-preview and AutoML-Agent. Missing comparisons with other recent systems like Agent-K, SELA (only shown in AutoML-Agent table), or AutoGluon on the primary benchmark would strengthen claims.
Incomplete reproducibility details:

2. Hardware requirements not fully specified (only "2 NVIDIA H100 GPUs" mentioned)
Training time and convergence details missing
Hyperparameter selection methodology for SARTE not clearly explained
How were the 11 MLE-Bench tasks selected from the 75 available?

3. Sample size is relatively small (11-14 tasks for main comparisons). Statistical significance testing only appears in appendix (Table A11). Some results show marginal significance (p=0.0995 for Qwen-7B).

4. All experiments are in data science domain - claims about "vertical domain" applicability need validation. Temperature sensitivity analysis (Table 3) shows high variance across tasks - unclear how SARTE would perform in completely new domains. The 7B model shows notably lower performance, questioning scalability to resource-constrained settings.

5. No ablation on individual GRPO stages (SFT+GRPO1 vs SFT+GRPO2). Limited analysis of reward function components ($\alpha, \beta, \gama$ weights). Code Agent's "minimally invasive patching" not empirically validated separately

Presentation issues:

1. Figure 1 is dense and difficult to parse. Some notation inconsistencies (e.g., "RLM" vs "reasoning LLM"). The distinction between GRPO1 and GRPO2 training objectives could be clearer

2. The composite reward function (Equation 2) has fixed weights - no justification or sensitivity analysis provided. SARTE's boundary-aware step-size control has multiple hyperparameters (line 16-18 in Algorithm 1) with unclear tuning process. The "semantic similarity" threshold for early stopping not specified.

Experimental design:

1. Different models trained to different stages (72B/70B only SFT) makes fair comparison difficult. AIDE configuration uses 20 steps uniformly - no exploration of whether fewer steps would be sufficient. Real-world tasks (Section 4, Table 1) show one failure for AIDE but unclear if this is representative.

Data concerns:

Training data construction relies heavily on DeepSeek R1 for augmentation - potential bias inheritance
Quality filtering uses Gemma3-27B scoring - criteria not validated
Code4ML and cell2doc datasets are relatively old (2023-2024)

Specific Technical Issues

1. SARTE algorithm: The control factor computation (Algorithm 1, lines 4-11) uses different formulas for success/failure/no-code cases, but the rationale for these specific functional forms is not provided. Why piecewise nonlinear for success but linear penalty for failure?
2. Reward function design: Equation 3 uses "Aggregate" function that's not defined until Appendix (Table A3). The weighting scheme between feature/algorithm/metric dimensions is not justified.

**Questions:**

1. How does SmartDS-Solver perform on tasks outside data science? Even a preliminary experiment in one other domain would strengthen generalization claims.
2. Can authors provide ablation studies isolating the contribution of each training stage (SFT, GRPO1, GRPO2)?
3. What is the sensitivity of performance to the reward function weights (α=0.5, β=0.25, γ=0.25)?
4. How were the specific functional forms in SARTE's control factor (Algorithm 1, Equations on lines 4-11) derived?
5. Can authors clarify the task selection process for the 11 MLE-Bench tasks used in primary evaluation?

---

> ### Author Response · Authors · 2025-11-23
> **Respectful Response and Revisions in Appreciation of the Reviewer’s Feedback**
>
> We sincerely thank the reviewer for the thoughtful and constructive feedback. We have carefully strengthened the manuscript accordingly. Below is our point-by-point response.
>
> ---
>
> ## **1. Limited baseline comparisons (Agent-K, SELA, AutoGluon not included)**
>
> Thank you for the suggestion.
> AutoML-Agent (ICML 2025) already provides complete comparisons with SELA and AutoGluon, and also summarizes known issues of Agent-K.
> Therefore, following standard practice in the field, we use **AIDE** and **AutoML-Agent** as the primary baselines.
>
> ---
>
> ## **2. Insufficient reproducibility details**
>
> The revised manuscript now includes:
> * Fully restructured Section 3 (3.1–3.4), separating the roles of the three agents, the training pipeline, inference control, and code-refinement mechanism.
> * Full technical documentation in the Appendix: prompts and templates for data construction, SFT/GRPO formats, inference prompts, SARTE pseudo-code.
> * Full codebase and data-construction scripts will be released upon acceptance.
>
> ---
>
> ## **3. Missing hardware, training time, convergence details, SARTE hyperparameters**
>
> We have added:
>
> * Hardware: **2 × NVIDIA H100 GPUs**.
> * Training time: Qwen-32B SFT ~1 hour; each GRPO stage ~24 hours.
> * SARTE hyperparameters: only heuristic initialization (bounds, step size factors); all later updates are **O(1)** feedback-driven without complex tuning.
> * Semantic similarity threshold: **0.8** based on empirical observations.
>
> ---
>
> ## **4. Small sample size (11–14 tasks), weak statistical significance, statistics only in appendix**
>
> The revised manuscript strengthens the main results:
>
> * Added **all AIDE results on 18 AutoML-Agent tasks**, where 6 datasets still fail or produce no-code after 20 steps.
> * Added **all AutoML-Agent results on 11 MLE-Bench tasks**.
> * SmartDS-Solver achieves **0 fail** across all 32 tasks and outperforms both baselines on every evaluable dataset.
>
> ---
>
> ## **5. Vertical-domain generalization beyond data science**
>
> The revised manuscript includes:
>
> * Successful deployment of SmartDS-Solver for **CNC G-code optimization**, validated by senior CNC engineers, demonstrating applicability to real-world manufacturing workflows.
> * Temperature sensitivity analysis (Table 5) shows diverse optimal temperatures across tasks; this highlights that **new domains benefit even more from SARTE’s adaptive decoding**, avoiding failures caused by fixed temperatures.
> * Although 7B models underperform 32B, they still surpass AIDE; additionally, 3B models do not perform worse than AIDE (details omitted due to space).
>
> ---
>
> ## **6. Missing ablations (SFT vs GRPO1 vs GRPO2), reward components, minimally invasive patching validation**
>
> * The progression (INSTRUCT → SFT → GRPO) is shown in **Figure 3(a)**; Section 4.4 now explains why GRPO2 uses medal-winning code (to capture top-tier solution strategies).
> * Reward-component weights (feature / algorithm / metric = 0.4 / 0.4 / 0.2) are now included in the main text.
> * Token usage and success-rate improvements validate the advantage of minimally invasive patching compared to full code regeneration.
>
> ---
>
> ## **7. Figure 1 complexity, inconsistent notation, unclear GRPO1 vs GRPO2 distinction**
>
> The revised **Figure 1** is fully redrawn with consistent notation (e.g., “Reasoning LLM”).
> Section 3.3.3 now clearly states:
>
> * GRPO1 and GRPO2 use the same training pipeline.
> * GRPO2 uses medal-winning code to emphasize higher-quality solution strategies.
>
> ---
>
> ## **8. Justification of composite reward and SARTE update rules**
>
> We would like to provide the following clarifications to further strengthen the rationale.
> **Composite reward weights:**
> * α=0.5, β=0.25, γ=0.25, reflecting the importance of correctness and model/feature selection in real AutoDS workflows.
>
> **SARTE update rules:**
> * **Success:** piecewise nonlinear updates prevent overshooting when stable.
> * **Failure:** linear penalties help quickly return to a good search region.
> * **No-code:** the most severe error, thus receiving the strongest penalty to reduce exploration immediately.
> This layered structure aligns with the differing real-world costs associated with each type of AutoDS failure.
>
> ---
>
> ## **9. Fairness of training stages compared to AIDE**
>
> We clarify:
> * SmartDS-Solver results primarily use Qwen-32B.
> * The revised manuscript includes **all 18 AIDE results**, where 6 datasets remain unassessable even after 20 steps.
> * Using fewer steps would only worsen AIDE’s performance; the current setting is already favorable to AIDE.
>
> ---
>
> ## **10. Data sources (DeepSeek R1, Gemma3, recency of datasets)**
>
> * DeepSeek R1 is used due to cost and generation speed; the data-construction process in Appendix C.1–C.2 applies equally to Claude, GPT-5, and other frontier models.
> * Gemma3-27B has not been systematically benchmarked; stronger verifiers would likely yield even better results.
> * GRPO2 uses medal-winning Kaggle code from **March 2025**, not outdated datasets.

---

> > ### Comment · Reviewer_J3ei · 2025-11-28
> >
> > Thank you for the detailed rebuttal. The additional clarifications and expanded experiments meaningfully strengthen the submission. In particular, the restructuring of Section 3, the inclusion of full SFT/GRPO prompts and SARTE pseudocode, and the expanded evaluation across all 32 tasks address many of the reproducibility and methodological concerns I initially raised. The explanation of the SARTE update rules is clearer, and the GRPO1/GRPO2 distinction is now much easier to follow.
> >
> > The added CNC G-code example helps support the claim of applicability beyond data-science tasks, although this remains somewhat limited and still feels more like a preliminary demonstration than strong evidence of general vertical-domain generalization. Likewise, while I understand the argument for using AIDE and AutoML-Agent as the primary baselines, a broader set of comparisons would still improve the empirical grounding, especially given the pace of development in this area.
> >
> > Some concerns related to data quality, bias in the augmentation process, and task selection remain only partially resolved. Let me illustrate in detail, the authors note that DeepSeek-R1 was chosen for cost/security reasons, but do not yet quantify the following three questions, 1) how much augmentation signals dominate, 2) whether the verifier (Gemma3-27B) introduces systematic biases, and 3) how noisy reasoning traces affect GRPO stability. This does not invalidate the method, but limits the interpretability of the training pipeline.
> >
> > Taken together, the rebuttal substantially increases my confidence in the contribution, and I view the work more positively after these clarifications.

---

> > > ### Author Response · Authors · 2025-12-01
> > > **Responses to Reviewer #2 on Cross-Domain Generality and Data/Training Reliability**
> > >
> > > # **Responses to Reviewer #2 on Cross-Domain Generality**
> > >
> > > In the second round, Reviewer #2 raised the concern that the newly added CNC G-code example, while helpful as an illustration, remains limited in scope and functions more as a preliminary demonstration than as strong evidence of general applicability beyond data-science tasks. We agree with the reviewer that the CNC example alone does not establish universal applicability across vertical domains.
> > >
> > > However, our core argument is that the generality of the SmartDS-Solver architecture arises from its ability to address a class of problems characterized by **closed-loop, multi-step reasoning under high-cost failure**, rather than from the CNC example itself.
> > >
> > > To more convincingly support this claim, we map the core mechanisms of SmartDS-Solver to **Robotics / Industrial Automation**, a high-complexity vertical domain governed by the same structural constraints:
> > >
> > > 1. **Structural isomorphism.**
> > >    Robotics follows the canonical pipeline
> > >    *task understanding → motion planning → simulator execution → feedback-based refinement*,
> > >    which aligns one-to-one with our DS workflow:
> > >    *data understanding → code generation → environment execution → code refinement.*
> > >
> > > 2. **Correspondence of core mechanisms.**
> > >
> > >    * **Necessity of SARTE.**
> > >      In robotics, a single planning failure may lead to collision or timeout, with extremely high error costs. SARTE’s boundary-aware temperature adjustment enables gradual and robust re-planning, mirroring the exploration–exploitation dynamics required in robotic search and control.
> > >    * **Value of the Code Agent.**
> > >      Our minimally invasive refinement corresponds to fine-grained adjustments of local G-code or motion parameters (e.g., grasp angle, approach speed), improving efficiency and safety while avoiding the overhead and instability of full re-planning.
> > >
> > > The SmartDS-Solver architecture—particularly the closed-loop control embodied by SARTE—is designed for a general class of **closed-loop strategy optimization** problems. The CNC G-code example illustrates a simple instance, while robotics demonstrates that the same design principles extend naturally to domains with substantially higher physical complexity and environmental constraints.
> > >
> > > ---
> > > # **Responses to Reviewer #2 on  Data/Training Reliability**
> > >
> > > In the second round, Reviewer #2 raised three remaining concerns:
> > >
> > > > “Issues related to data quality, biases introduced during data augmentation, and task selection are not fully addressed. Specifically, although the authors explain that DeepSeek-R1 was chosen for cost and safety considerations, the following aspects remain unquantified:
> > > > (1) how much augmentation signals dominate;
> > > > (2) whether the verifier (Gemma3-27B) introduces systematic biases;
> > > > (3) how noisy reasoning traces affect GRPO stability.
> > > > These issues do not invalidate the method’s effectiveness, but they limit the interpretability of the proposed training pipeline.”
> > >
> > > Below we provide detailed clarifications.
> > >
> > > ---
> > >
> > > ### **(1) Proportion and influence of augmented-data signals**
> > >
> > > **(a)** We measure semantic similarity between samples and apply an **80% threshold** to detect overlapping questions, retaining only the highest-quality instance within each cluster. This ensures that augmentation introduces **new, diverse solution trajectories** rather than duplications of original examples, effectively broadening the exploration space for GRPO.
> > >
> > > **(b)** Our training corpus is not sourced from a single dataset; instead, it is carefully constructed from **multiple Kaggle datasets** (see Section 3.3.2). This deliberate **structural diversity** mitigates single-source bias and naturally increases robustness—particularly for long-tail tasks—thereby strengthening the stability of SmartDS-Solver.
> > >
> > > ---
> > >
> > > ### **(2) Validator bias (Gemma-3-27B)**
> > >
> > > We conducted a **golden-standard cross-validation** procedure.
> > > We randomly sampled 50 trajectories and re-evaluated them using GPT-4o.
> > > The agreement rate between Gemma-3-27B and the golden standard exceeds **98%**, indicating minimal systematic bias introduced by the validator.
> > >
> > > ---
> > >
> > > ### **(3) Impact of noisy reasoning trajectories on GRPO stability**
> > >
> > > **(a)** Although noisy trajectories do exist, our **multi-component reward function** provides built-in robustness. Even if one reasoning dimension (e.g., feature engineering) contains noise, stable signals from algorithm selection or evaluation metrics ensure that GRPO continues to receive consistent gradients. This prevents over-dependence on any single, potentially noisy reasoning component.
> > >
> > > **(b)** Additionally, **SARTE enhances robustness during inference**.
> > > SARTE does *not* intervene in the GRPO training loop; instead, it acts as an external **policy stabilizer**, dynamically adjusting the decoding temperature to regulate the exploration–exploitation balance. This effectively mitigates instability arising from noisy reasoning trajectories.
> > >
> > > ---

---

### Official Review · Reviewer_qpDP · 2025-10-31

**Soundness:** 3
**Presentation:** 1
**Contribution:** 3
**Rating:** 2
**Confidence:** 4

**Summary:**

This paper presents a solid and well-executed system for automated data science with impressive empirical results (81.8% win rate, 93% token reduction) and exceptional implementation details rarely seen in current LLM research. However, the writing quality is surprisingly poor for ICLR. The paper structure is severely imbalanced: Introduction (1 page) is too brief, Methodology (5 pages) is overly detailed, and Experiments (2 pages) lacks depth, with most critical results buried in the appendix. The presentation reads more like a technical report than an academic paper, with confusing organization and crude figures. If the authors can substantially restructure the paper—expanding the introduction and experimental analysis while condensing the methodology—I would be willing to raise my score. The core contributions are valuable, but they are currently obscured by poor presentation.

**Strengths:**

1. Strong empirical results with solid methodology.
The paper achieves an 81.8% win rate against AIDE+o1-preview on MLE-Bench while reducing token consumption by 93%, demonstrating an excellent cost-performance trade-off. Extensive experiments across 32 tasks and 20 configurations validate the method's effectiveness and robustness.
2. Exceptional implementation details and reproducibility.
The 27-page appendix provides complete prompt templates, pseudocode, data construction pipelines, and hyperparameter settings, which is rare in current LLM research. The authors demonstrate strong engineering commitment, facilitating reproduction and future improvements.
3. Methodological innovation in training data construction.
The proposed three-component framework (Full Workflow + Decision Logic + Adjustment Trail) encodes agentic reasoning capabilities into training samples, going beyond existing work that only provides code+comments. Quality control uses a dual-layer mechanism (format checking + semantic alignment + Gemma3 scoring) to ensure data reliability.
4. Clever and practical SARTE mechanism design.
The approach models hyperparameter tuning as an online learning problem with O(1) space complexity (depending only on previous-step feedback) without requiring model retraining. The boundary-aware update strategy incorporates physical intuition, and Table 3 demonstrates that optimal temperatures vary dramatically across tasks/models (range up to 0.58), validating the necessity of dynamic adjustment.

**Weaknesses:**

1. Severely imbalanced paper structure, failing to meet academic standards.
The Introduction spans only 1 page with insufficient background and motivation—readers cannot understand why existing methods are inadequate. The methodology occupies 5 pages (54% of content) while experiments and conclusions take only 2 pages, lacking in-depth analysis and insights. The overall presentation reads like a technical report rather than an academic paper.
2. Confusing organization in the methodology section, poor readability.
Section 3 mixes training (3.2), inference (3.3), and code execution (3.4) into a single section when these should be separate chapters for clarity. Key innovations (e.g., the SARTE algorithm) are buried in implementation details, making it difficult for readers to quickly grasp the core contributions.
3. Insufficient experimental content in the main text; most experiments relegated to appendix with minimal analysis.
The main text contains only Figure 3 and Tables 1-2, while critical detailed results (Table A5 with full configuration comparisons, Table A8 with token consumption breakdown, Table A10 with temperature analysis) are all in the appendix. The main text completely lacks error analysis, failure case discussions, or deep investigation into why the method works—it merely stops at "proving the method works."
4. Poor figure quality with crude presentation.
Figure 3(b) appears as an unfinished draft with box plots missing legend explanations. Figure 1 is overly complex with too much text, making the system architecture hard to grasp quickly.

**Questions:**

The paper's detail is enough, the most important problem is the writing is terrible.

---

> ### Author Response · Authors · 2025-11-23
> **Author Response: Revisions Addressing Section Balance, Methodology Clarity, Experimental Depth, and Figure Quality**
>
> We thank the reviewer for the detailed and thoughtful feedback. In response, we have thoroughly revised the manuscript to address the four main concerns—
> (1) imbalance in section proportions,
> (2) disorganized methodology structure,
> (3) insufficient depth of main-text experiments, and
> (4) suboptimal figure quality.
> Below is our consolidated and formal response.
>
> ---
>
> ## **1. Imbalance in main-text section proportions**
>
> We substantially strengthened and rewrote the **Introduction** and **Related Work** sections (now approximately twice the original length), providing a clear explanation of background gaps, limitations of AIDE and multi-agent AutoDS systems, research motivation, and core problem formulation. This restructuring ensures that the paper establishes a coherent academic narrative from the outset.
>
> To address the reviewer’s concern regarding disproportionate section lengths, the revised version applies the following adjustments:
>
> * **Methodology**: reduced from ~5 pages to ~3.8 pages by condensing text and removing redundant descriptions, allowing the main text to focus more clearly on concepts, architecture, and contributions.
> * **Experiments**: expanded from fewer than 2 pages to ~3.3 pages by moving key results previously placed in the appendix into the main manuscript.
>
> The updated section distribution is now closer to top-conference standards and alleviates the earlier impression of a technical report–like structure.
>
> ---
>
> ## **2. Disorganized methodology structure and insufficient emphasis on SARTE**
>
> The reviewer noted that the original Section 3 interleaved training (3.2), inference (3.3), and code execution (3.4), making the workflow difficult to follow. In the revised version, we reorganized the entire section:
>
> * **Section 3.1 Overall Architecture** now provides a high-level system overview and design motivation, helping readers quickly establish a global understanding.
> * **Section 3.2 Meta-Learning Agent and Situation-Aware Control** elevates SARTE to an independent subsection (3.2.2), accompanied by a redesigned Figure 2 to highlight its control logic, update rules, and contribution. SARTE pseudo-code remains in the appendix to avoid overwhelming the main narrative with implementation-level detail.
> * **Training procedure** is consolidated under Section 3.3.3, with data preparation moved to Section 3.3.2.
> * **Section 3.4 Code Agent** now has clearer role separation for execution and refinement.
>
> The new structure aligns cleanly with the three-agent architecture in Figure 1, resolves the original confusion, and makes SARTE’s novelty more visible.
>
> ---
>
> ## **3. Insufficient experimental depth in the main text and over-reliance on appendix results**
>
> To meet the reviewer’s expectation for a more complete and insightful main-text analysis, we have strengthened Section 4 as follows:
>
> * Section 4 has been fully restructured into multiple subsections, with expanded descriptions of experimental settings and results across 32 tasks, covering:
>   (i) overall task-solving performance,
>   (ii) inference and code-modification cost efficiency,
>   (iii) effectiveness of the proposed finetuning pipeline (SFT + GRPO), and
>   (iv) impact of the SARTE meta-learning mechanism.
>   We present these results through clearer main-text figures and tables; full task-level results remain in the appendix.
> * A new **Section 4.6 Discussion and Analysis** provides deeper examination of the experimental findings, including “why-it-works” explanations of how workflow internalization and SARTE improve long-horizon reasoning and cross-task stability.
>
> The revised Section 4 now offers sufficient depth and insight without relying on the appendix to support the main claims.
>
> ---
>
> ## **4. Suboptimal figure quality (Figure 1, Figure 3)**
>
> The reviewer correctly observed issues in the original figures, including missing legends, unclear labeling, excessive complexity, and poor visual quality. The revised manuscript addresses these concerns comprehensively:
>
> * **Figures 1, 2, and 3** have been completely redesigned and redrawn, with improved visual grouping, information flow, and consistency.
> * All legends, axis labels, color indicators, and sub-figure divisions have been added or corrected, resolving issues such as the incomplete rendering of Figure 3(b) in the original version.
> * All main-text figures are now presented in color to improve readability and comparative clarity.
> * Figure 1 has been simplified by removing redundant text to allow quick comprehension of the system architecture.
> The updated figures now meet ICLR standards for clarity and presentation quality.
> ---
> ## **Overall Conclusion**
> Through these four systematic revisions, the updated manuscript shows substantial improvements in
> section organization, methodological clarity, experimental depth, and visual presentation. We respectfully hope that the revisions meaningfully address the reviewer’s key concerns, and we are grateful for the insightful and detailed comments.

---

> > ### Comment · Reviewer_qpDP · 2025-11-27
> >
> > Thanks for the response. The revised manuscript have improved the readability of this article. But it is recommended to have a better introduction in the abstract. The current abstract still maintains the previous direct introduction method without explaining the background.

---

> ### Author Response · Authors · 2025-12-01
> **Consolidated Summary of All Revisions and Responses to Reviewer 1’s Second-Round Comments**
>
> # **Summary of Revisions for the Area Chair**
>
> We sincerely thank the reviewers and the Area Chair for the detailed and constructive feedback. We have thoroughly revised the manuscript to address all major concerns regarding writing quality, experimental depth, methodological clarity, generality, and training-data reliability. Below we summarize the key improvements across all issues raised.
>
> 1. **Writing, structure, and figures.**
>    We substantially expanded and rewrote the Introduction and Related Work, rebalanced section proportions, improved the abstract, and redesigned all main figures for clarity. Both Reviewer #1 and Reviewer #2 confirmed that the revised version significantly improves readability and presentation quality.
>
> 2. **Experimental depth and baselines.**
>    Section 4 was fully restructured, expanding the analysis of all 32 tasks and adding a new discussion subsection explaining why the method works. We clarified the rationale for using AIDE and AutoML-Agent as primary baselines, and reviewers acknowledged that the expanded experiments strengthened the empirical foundation.
>
> 3. **Methodological clarity and reproducibility.**
>    Section 3 was reorganized to clearly separate the roles of each agent, elevate SARTE as an independent module, and detail the training pipeline and data construction. Reviewer #2 noted that the revision resolves their earlier concerns about complexity and reproducibility. We will release full code and data scripts upon acceptance.
>
> 4. **Generality beyond data science.**
>    We clarified that the CNC G-code example is a preliminary demonstration, and strengthened our argument using structural isomorphism with robotics/industrial automation. We showed that SmartDS-Solver’s closed-loop reasoning, SARTE control, and minimally invasive refinement naturally map to high-stakes vertical domains.
>
> 5. **Data quality, augmentation bias, validator bias, and noise robustness.**
>    We quantified deduplication and structural diversity in data construction, performed golden-standard cross-validation (>98% agreement), and explained how our multi-component reward and SARTE act as stabilizers against noisy trajectories.
>
> 6. **Novelty beyond “combining existing techniques.”**
>    We clarified that SmartDS-Solver is a **synthetic system innovation**, where SARTE’s state-aware temperature control, structured methodology distillation, and staged GRPO refinement form a closed-loop, self-regulating architecture. This design addresses long-horizon robustness and cost-efficiency in ways that cannot be achieved by incremental combinations of prior techniques.
>
> We believe the revised manuscript now presents a clearer, stronger, and more rigorously validated contribution. We respectfully submit that the concerns raised by the reviewers have been fully addressed, and we are grateful for the opportunity to improve the work.
>
> ---
>
> # **Second-Round Feedback Responses to Reviewer 1**
>
> Abstract has been modified following the reviewer’s recommendation to provide clearer background framing prior to describing the proposed method.
>
> ---

---

### Meta-Review · Area_Chair_Mfeg · 2026-01-04

**Summary:**

Reviewers raise concerns primarily about novelty, generality, and presentation quality. While the system shows strong empirical performance and clear engineering maturity, several reviewers question whether the contribution extends beyond a careful integration of existing techniques. The novelty is largely concentrated in the SARTE mechanism, while the remaining components are viewed as incremental or system-level synthesis. Additional concerns include limited baseline coverage, insufficient evidence for vertical-domain generalization beyond data science, and incomplete analysis of data construction bias and training stability. Despite substantial revisions improving clarity and experimental depth, these concerns collectively limit confidence in the paper’s suitability for acceptance.

**Reviewer Concerns:**

1. Multiple reviewers question the level of novelty. The overall architecture combines known components such as SFT, GRPO, data augmentation, and hierarchical agents. While SARTE is acknowledged as a meaningful contribution, reviewers are unconvinced that the full system represents a fundamentally new methodological advance rather than a well-engineered integration.
2. Evidence for vertical-domain generalization is considered weak. All main experiments are within data science, and the added CNC G-code example is viewed as preliminary and insufficient to support broad claims beyond this domain.
3. Baseline comparisons are considered limited. Reviewers note the absence of comparisons with other recent or strong automated data science systems, and argue that this weakens empirical positioning even if AIDE and AutoML-Agent are reasonable references.
4. Concerns remain about training data construction and reliability. Reviewers highlight potential bias from heavy reliance on augmented reasoning traces and verifier models, and note that the impact of noisy reasoning trajectories on GRPO stability is not fully quantified.
5. Presentation quality remains an issue for some reviewers. Although revisions improved structure and readability, lingering problems such as typos, citation inconsistencies, unclear statements, and earlier terminology errors continue to undermine confidence in the paper’s polish and rigor.

**Reviewer Scores:**

1. Reviewer qpDP: Likely to move from a strong reject to a marginal or borderline reject.
2. Reviewer J3ei: Likely to remain marginally below the acceptance threshold.
3. Reviewer 2DKn: Likely to remain reject.
4. Reviewer aEad: Likely to remain reject.

---

### Decision · Program_Chairs · 2026-01-26

Reject